# Bio-inspired Working Memory for Online Auditory Pattern Drift Detection

## Abstract

Recent advances in Audio Language Models (ALMs) have attracted unprecedented attention. However, transformer-based ALMs face challenges in long-form audio understanding due to inefficient attention allocation. To address this, we introduce a biologically inspired working memory module, BioWM (Bio-inspired Working Memory), which leverages unsupervised online drift detection as an adaptive attention allocation strategy. BioWM detects auditory pattern drifts by monitoring energy fluctuations induced by spatio-temporal shifts, enabling the model to focus on salient changes. The BioWM does not require long-term historical data or offline pretraining; instead, it adapts online with only a few steps of threshold adjustment. Our approach captures novel events while remaining robust to transient perturbations. Furthermore, BioWM exhibits oscillatory frequency-band dynamics that resemble cortical activity during working memory tasks, thereby strengthening its biological plausibility. We present comprehensive experiments demonstrating the effectiveness of BioWM and provide visualizations of its evolving internal states to highlight both performance gains and interpretability.

## 1 Introduction

Recent Audio Language Models (ALMs) have advanced audio understanding (Chu et al., 2023; 2024) but remain too computationally demanding for real-time long-form analysis (Hou et al., 2024; Guo et al., 2025b). Their limits in attention and resources call for more efficient strategies than uniform sequence processing. Inspired by human selective attention (Kauramäki et al., 2007; Huang & Elhilali, 2020), models should dynamically allocate computation to salient auditory events while maintaining stable background representations (Kasten et al., 2024).

A promising approach to this challenge involves detecting auditory temporal-pattern salience—the online identification of deviations from expected regularity in sound sequences. Such deviations may manifest as novel sound events, or categorical changes in sound sources. However, in real-world applications, offline training approaches incur prohibitive costs due to expensive data collection and relabeling requirements. This necessitates online, unsupervised methods that can adapt without extensive historical data or retraining phases (Wan et al., 2024; Chan et al., 2025).

Current AI approaches to online unsupervised drift detection include statistical-based methods (Rabanser et al., 2019; Chan et al., 2025) and contrastive learning techniques (Wan et al., 2024). However, these methods show limited effectiveness in scenarios containing diverse acoustic events and require long-term historical data (Wan et al., 2024) or substantial offline pretraining datasets (Greco et al., 2025). These limitations prevent them from distinguishing between meaningful pattern changes and natural variations within ongoing audio streams, such as transient pauses in speech or music. Current methods struggle with this distinction, limiting their utility as attention-gating mechanisms for higher-level processing.

These challenges motivate seeking biological inspiration from neuroscientific approaches to auditory processing. Research indicates that auditory working memory maintains different sound events as discrete neural attractors (Brennan & Proekt, 2023), while novel events activate specific neural patterns (Huang & Elhilali, 2020) within this working memory system, typically associated with enhanced gamma-band activity (Albouy et al., 2017; Bonetti et al., 2024). These biological mecha-

nisms suggest that oscillatory working memory representations may provide a principled foundation for computational drift detection that balances stability with sensitivity to meaningful changes.

This paper introduces NAACA (**N**euro**A**uditory **A**ttentive **C**ognitive **A**rchitecture), a bottom-up framework that simulates auditory processing pathways (Huang & Elhilali, 2020). Our system processes audio streams through a multi-stage pipeline: incoming audio is first encoded into semantic events, then passed through a temporal modulator that maps these events to the core component BioWM (**Bio**logically-inspired **W**orking **M**emory), and finally uses energy detection to determine when information should be transmitted to higher cognition modules.

The core BioWM component operates with two types of neurons: primary neurons that serve as main information carriers, and velocity neurons that act as spatial modulators (Watakabe et al., 2023). The BioWM grid-like spatial coupling is governed by wave equations, so that these grids exhibit frequency selectivity, which ensures high discriminability between different attractors and facilitates change detection. Meanwhile, the coupling also creates short-term memory effects where oscillations persist even after events cease, making the system robust to transient changes such as pauses.

Our contributions are threefold:

- We propose NAACA, a bio-inspired framework with BioWM as its core component, enabling unsupervised online auditory pattern drift detection without requiring historical data or training phases, addressing the computational bottleneck of existing methods.

- Our oscillatory, wave-based BioWM design exhibits biologically plausible dynamics including gamma-band activity enhancement during novel event detection, while achieving computational efficiency through spatial-frequency separation that reduces false positives from natural audio variations.

- We demonstrate NAACA's ability to identify three distinct types of drift (novel event onset, transient pause robustness, and subcategory-level changes) in urban soundscapes with less false positive detection compared to traditional similarity-based approaches.

## 2 METHODS

### 2.1 NEUROAUDITORY ATTENTIVE COGNITIVE ARCHITECTURE (NAACA)

Our framework follows a multi-stage processing pipeline (Fig. 1 and Algorithm B.1). Incoming audio streams are segmented into short, overlapping windows $\mathbf{x}_t$, which are encoded into event-level probability vectors $\mathbf{p}_t = \mathrm{Enc}(\mathbf{x}_t)$ by a pretrained encoder $\mathrm{Enc}(\cdot)$.

These probabilities are transformed into oscillatory drive signals through a predefined modulator $\mathcal{M}(t)$. Specifically, each probability dimension is assigned a unique carrier frequency, represented as a sine wave, while the corresponding probability value modulates its amplitude. As illustrated in Fig. 1, three events (*speech*, *traffic*, and *music*) produce event-level probabilities across five sliding windows, which are then mapped to their respective sinusoidal drive signals at distinct frequencies. Formally, the modulated source term for event $i$ is defined as

$$S_i(x,t) \;=\; a_i(t)\,\sin(\omega_i t)\,\mathbf{1}_{\Omega_i}(x), \qquad \omega_i = 2\pi f_i, \;\; a_i(t) \in [0,1], \tag{1}$$

where $a_i(t)$ denotes the instantaneous amplitude $\mathbf{p}_t$ given by the encoder probability for event $i$ at time $t$, $f_i$ is the unique carrier frequency assigned to event $i$, $\omega_i = 2\pi f_i$ is the corresponding angular frequency, $\Omega_i \subset \{1,\ldots,G\} \times \{1,\ldots,G\}$ represents the spatial allocation (parcel) within the 2D recurrent network dedicated to event $i$. Each $\Omega_i$ acts as an **attractor** dedicated to a single event, and $\mathbf{1}_{\Omega_i}(x)$ is the indicator function that activates the assigned spatial region $\Omega_i$. This spatial embedding ensures that bio-inspired wave dynamics within the BioWM can store, propagate, and dissipate event-specific information in a structured manner.

We monitor changes in the BioWM's internal energy profile against an adaptive threshold $T_{\mathrm{adapt}}$ to detect pattern drift. The $T_{\mathrm{adapt}}$) is computed using an energy-based approach:

$$T_{\mathrm{adapt}} = \mu + 2\sigma\big(1 + \alpha \cdot \mathrm{trend}\big), \tag{2}$$

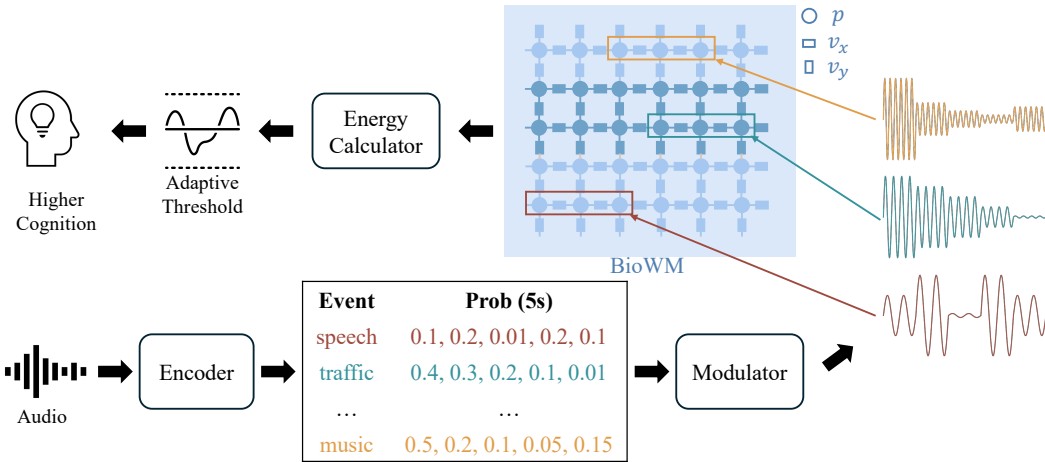

Figure 1: **Overview of the NAACA (NeuroAuditory Attentive Cognitive Architecture).** Audio is segmented into sliding windows and mapped by a pretrained encoder to event probabilities that drive frequency-specific oscillatory inputs on BioWM grids. BioWM is a 2D neural network with primary ($p$) and velocity ($v$) neurons, parameterized by wave speed $c$ and damping $k$, where $c$ is stripe-shaped binary (dark/light blue). Drift is detected via energy fluctuations against an adaptive threshold, and events are forwarded to a higher-level cognition module for semantic interpretation.

where $\mu$ and $\sigma$ are the running mean and standard deviation, and the trend factor adjusts for temporal patterns in the data. The final detection decision employs persistence filtering to ensure robust drift detection while minimizing false positives. The whole procession is shown in Algorithm B.2.

Detected drift patterns are then forwarded to a higher-level cognitive module responsible for identifying the nature of the drift.

## 2.2 BioWM Formulation

BioWM is a 2D recurrent field model defined on a $G \times G$ lattice. Its hidden state $\mathbf{h}_t \in \mathbb{R}^{G^2}$ stores a primary field $p(x, y, t)$ whose activity propagates via a structured spatial operator $\mathcal{S}(\cdot)$ with wave-like coupling. The update combines (i) temporal recurrence through $\mathbf{h}_{t-1}$ for memory and (ii) spatial recurrence through $\mathcal{S}(\cdot)$ for lateral propagation. An intermediate velocity-like variable $\mathbf{v}(x, y, t) = (v_x, v_y)$ mediates directional flow. Parameters $c$, $k^p$, and $k^v$ control propagation speed and damping. This bio-inspired design is analogous to membrane-potential storage ($p$) and axonal/dendritic transport ($\mathbf{v}$) in cortical sheet.

### 2.2.1 Constructing the Structured Spatial Operator $\mathcal{S}(\cdot)$ of BioWM

**Wave System Foundations.** We consider the BioWM as a two-dimensional oscillatory system governed by a damped wave equation in the first-order velocity–pressure formulation:

$$\frac{\partial p}{\partial t} + k^p(x, y)\, p = -c^2(x, y)\, \nabla \cdot \mathbf{v} + S(x, y, t),$$

$$\frac{\partial \mathbf{v}}{\partial t} + k^v(x, y)\, \mathbf{v} = -\nabla p,$$

$$(3)$$

where $c(x, y)$ is the spatially varying wave speed (time-independent), $k^p(x, y)$ and $k^v(x, y)$ are pressure and velocity damping coefficients (time-independent), and $S(x, y, t)$ is the input excitation from event-level probabilities given by Eq. 1, and $\nabla = \begin{bmatrix} \partial/\partial x \\ \partial/\partial y \end{bmatrix}$ is the gradient operator, and $\nabla \cdot \mathbf{v} = \partial v_x/\partial x + \partial v_y/\partial y$ is the divergence operator.

**Discrete Formulation.** Discretizing Eqs. 3 over a two-dimensional lattice yields:

$$p(x, y, t + \Delta t) = \left[ 1 - \Delta t\, k^p(x, y) \right] p(x, y, t) - \Delta t\, c^2(x, y)\, \nabla \cdot \mathbf{v}(x, y, t) + \Delta t\, S(x, y, t),$$

$$\mathbf{v}(x, y, t + \Delta t) = \left[ 1 - \Delta t\, k^v(x, y) \right] \mathbf{v}(x, y, t) - \Delta t\, \nabla p(x, y, t), \tag{4}$$

where $\Delta t$ is the time step.

This discrete update system admits oscillatory solutions at each grid point, whose behavior can be characterized in terms of local eigenfrequencies, as stated by Theorem 1.

**Theorem 1** (Local Eigenfrequency Formula)**.** *At each grid location $(i, j)$, the discretized system admits a local eigenfrequency*

$$f_{i,j} = \frac{1}{\pi \Delta t} \tan^{-1}\left( \frac{2c(x, y)\Delta t \sqrt{\xi_x^2 + \xi_y^2}}{\sqrt{(1 + \Delta t k^p(x, y))(1 + \Delta t k^v(x, y))}\left( \frac{1}{1 + \Delta t k^p(x,y)} + \frac{1}{1 + \Delta t k^v(x,y)} \right)} \right), \tag{5}$$

*where $f_{i,j}$ is the eigenfrequency at spatial coordinate $(x, y)$ corresponding to grid index $(i, j)$, $\xi_x$ and $\xi_y$ are spatial frequency components, and all other parameters retain their definitions from the discrete wave system.*

*Proof.* See Appendix C. □

**System Energy Measurement.** Different spatial locations are associated with distinct attractors corresponding to different sound events. Since each event is characterized by its own modulatory input frequency, the local wave speed $c(x, y)$ must also vary across space. This spatial dependence ensures that the eigenfrequency structure reflects the diversity of sound-driven dynamics in the system.

However, while local eigenmodes are essential for modeling event-specific resonances, pattern drift detection and memory-related computations require more than analyzing these modes in isolation. Instead, the collective behavior of the system must be captured in terms of a global state variable. To this end, we define the total system energy, which aggregates pressure and velocity contributions across the lattice. This energy-based representation not only reflects the ongoing dynamics of the BioWM but also forms the key computational signal for detection and optimality analysis.

The total energy of the BioWM system in discrete form is

$$E(t) = \frac{1}{2} \sum_{i,j} \left[ p_{i,j}^2(t) + v_{x,i,j}^2(t) + v_{y,i,j}^2(t) \right], \tag{6}$$

where the terms correspond to kinetic energy due to coupling, potential energy due to stiffness. For the purpose of analyzing and designing the internal structure and parameters, we approximate the 2D lattice as a continuous medium. The energy then becomes

$$E(t) = \iint \left[ \frac{1}{2} p^2(x, y, t) + \frac{1}{2} v_x^2(x, y, t) + \frac{1}{2} v_y^2(x, y, t) \right] dx, dy, \tag{7}$$

which will be used in the following calculations and theorem proofs as the basis for sensitivity and optimality analyses.

### 2.2.2 Topological Organization with High Sensitivity to Drift

To analyze the role of topological organization and its sensitivity to drift, we first reformulate the governing first-order velocity–pressure system in a more compact representation in Theorem 2. This reformulation exposes the effective damping and restoring mechanisms and serves as a foundation for later connecting topological behavior with energy dynamics.

**Theorem 2** (Equivalence of First-Order System to Second-Order Damped Wave Equation)**.** *The first-order velocity–pressure system Eqs . 3 is equivalent to the second-order damped wave equation*

$$\frac{\partial^2 p}{\partial t^2} + (k^p + k^v)\frac{\partial p}{\partial t} + k^v k^p \cdot p = c^2 \nabla^2 p + \left( k^v + \frac{\partial}{\partial t} \right) S, \tag{8}$$

*with effective damping coefficient $\gamma = k^p + k^v$, restoring force coefficient $\mu = k^v k^p$, and modified source term $S_{eff} = \left( k^v + \frac{\partial}{\partial t} \right) S.$*

*Proof.* See Appendix D. □

Building on this structural equivalence, we next derive the explicit energy evolution law, which highlights how the wave speed $c$ governs energy redistribution and thereby influences stability and drift sensitivity.

**Theorem 3** (Energy Evolution). *When $p(x, y, t)$ and $v_x, v_y$ are governed by Eqs. 3 under periodic boundaries. The energy evolution is given by*

$$\frac{dE}{dt} = -\iint \left[k^p p^2 + k^v (v_x^2 + v_y^2)\right] dx\, dy - \iint (c^2 - 1)p(\partial_x v_x + \partial_y v_y)\, dx\, dy + \iint pS\, dx\, dy. \quad (9)$$

*Proof.* See Appendix E. □

To analyze the best wave speed distribution pattern, we define the *sensitivity of the energy change* to input perturbations $\delta S$ as $\delta\left(\frac{dE}{dt}\right) = \iint p \cdot \delta S\, dx\, dy$.

The pressure field $p(x, y, t)$ is determined by the system with energy redistribution effects, which can amplify oscillations and spatial gradients. As a result, larger values of $p(x, y, t)$ lead to higher sensitivity of the energy evolution to input fluctuations.

We next ask: what distribution of $c(x, y)$ best enhances the redistribution of $p(x, y, t)$, so that the system energy becomes maximally sensitive to pattern changes? The following Theorem 4 shows that binary-valued speed fields achieve this optimal sensitivity.

**Theorem 4** (Binary Contrast Optimality). *Under fixed mean and contrast (total variation) constraints on the wave speed field $c(x, y)$, binary-valued distributions maximize system response measures relevant to sensitivity and scattering.*

*Proof.* We establish the claim by tracing how binary contrasts affect three complementary mechanisms, each formalized in Appendix F. First, sharp transitions in the wave-speed field maximize interface reflections, yielding the strongest boundary-driven amplification (Lemma 1). Second, under a fixed total-variation (contrast) budget, concentrating contrast on fewer interfaces achieves maximal transfer efficiency; extreme (binary) allocations dominate any smooth or uniformly spread contrast (Lemma 2). Finally, these effects jointly amplify the pressure field and thus the energy-rate sensitivity to source perturbations, maximizing detection responsiveness (Lemma 3). Taken together, these mechanisms imply that binary-valued speed distributions maximize the relevant system response measures, which proves the theorem. □

Having established the benefit of binary contrasts in $c(x, y)$, we now examine which geometric arrangement of these values is most effective. Among possible layouts, striped patterns not only yield strong coupling but are also the most practical to realize.

**Theorem 5** (Striped Pattern Justification). *Among binary-valued configurations with periodic boundaries and fixed mean/amplitude, striped patterns (unidirectional variation) maximize total modal coupling strength and targeted mode-pair coupling.*

*Proof.* By Theorem 4, binary-valued speed fields maximize system response because sharp low–high $c$ interfaces yield the strongest sensitivity. Building on this, Lemma 4 shows that modal interactions are governed by the Fourier spectrum of the contrast pattern. Concentrating the contrast energy into few dominant spectral components therefore ensures maximal global coupling, as established in Lemma 5. A striped binary arrangement achieves precisely this by aligning all low–high $c$ boundaries in one direction, so that their contributions add coherently.

Lemma 6 further shows that for a target mode-pair $(m, n)$ and $(m, n+q_0)$, the coupling is maximized when the stripe period matches the modal separation, i.e., when the repeated low–high $c$ alternations resonate with the modal difference. Thus, striped patterns simultaneously maximize both global coupling (via energy concentration) and local targeted coupling (via period matching), establishing the claimed optimality. In terms of drift detection, strong global coupling enhances overall sensitivity to pattern changes, while strong local coupling ensures precise detection of specific modal shifts. □

## 3 RESULTS

### 3.1 EXPERIMENTAL SETTING

**Temporal input windows.** To capture auditory sensory input, we apply a sliding window of $4\,\mathrm{s}$ with a stride of $1\,\mathrm{s}$. Each windowed segment is processed independently through the framework, providing temporal continuity while preserving overlap across successive inputs. In addition, an *attention window* of $15\,\mathrm{s}$ is maintained to aggregate information over longer temporal spans. This attention window is forwarded to the higher cognition layer for contextual reasoning.

**Backbone feature extraction.** We adopt PANN (Kong et al., 2020) as the backbone feature extractor to obtain representations of audio segments. PANN produces a 2048-dimensional embedding vector and a classifier that outputs logits over 527 audio event classes. The logits are passed through a sigmoid activation to obtain clipwise probabilities in the range $[0, 1]$. These probabilities are used as amplitude modulators, controlling the gain of corresponding sine wave carriers assigned to each embedding dimension.

**BioWM model configuration.** The BioWM model employed in our experiments follows a 2D architecture with inputs represented on a $64 \times 64$ grid. The main hyperparameters are set as follows: $dt = 0.01$, $dx = 1$, and $k_o = k_p = 10$. The spatially-varying wave speed parameter $c$ follows the BioWM eigenfrequency formula (Theorem 1) with target frequencies linearly distributed across PANN dimensions within the range 50–1200 Hz. This yields a characteristic binary striped distribution with wave speeds clustering around $c \approx 0.1$ and $c \approx 70$, consistent with the theoretical optimality results of Theorems 4 and 5. The detailed calculation procedure is provided in Appendix H.

**Higher cognition layer.** For higher-level interpretation, we incorporate Audio Qwen (Chu et al., 2023) as the higher cognition layer. Specifically, Audio Qwen generates descriptive drift outputs when the BioWM model detects auditory pattern shifts.

### 3.2 DATASET AND MEASUREMENTS

**Dataset.** To evaluate our model under realistic auditory conditions, we use the *Urban Soundscapes of the World (USoW)* dataset (De Coensel et al., 2023; 2017). USoW was recorded at carefully selected urban locations across multiple cities worldwide and provides high-quality 4-channel ambisonics audio. For our experiments, we extract 1-minute audio segments from the recordings.

Compared to conventional environmental sound datasets, USoW is well suited for studying auditory pattern drift. It spans diverse anthropogenic and natural sound events, capturing dynamics of human auditory attention. Unlike prior audio-visual saliency tasks that emphasize localized events (e.g., instrument or speaker changes within a scene) (Liu et al., 2024; Guo et al., 2025a), USoW models scene-level variability across entire soundscapes, making it more appropriate for evaluating strategies that detect and track global auditory context shifts. The full comparision table is in Appendix M.

**Measurements.** As online unsupervised auditory pattern drift remains largely unexplored, no benchmark dataset or ground-truth labels are currently available for this task. We therefore evaluate our approach using representative examples and quantitative proxies. Specifically, we compute the *energy* metric from BioWM as our primary indicator and compare it against a baseline that measures cosine similarity between embeddings. Sensitivity to drift is assessed by reporting the fraction of detected drifts over all time steps, compared across samples and trials. Finally, we present a computational efficiency analysis in Appendix L to demonstrate the system's real-time feasibility.

### 3.3 ILLUSTRATIVE CASES OF AUDITORY PATTERN DRIFT

To qualitatively assess the behavior of our framework, we highlight three illustrative cases of auditory pattern drift. Specifically, we consider: (1) the appearance of a novel sound event that shifts the auditory context, (2) robustness to transient silences or pauses in ongoing sound streams, and (3)

sensitivity to subcategory-level substitutions within a broader sound category (e.g., different types of musical instruments within the "music" class). These cases provide concrete insights into how the BioWM detects and distinguishes different forms of drift beyond low-level acoustic fluctuations.

### 3.3.1 APPEARANCE OF A NOVEL SOUND EVENT

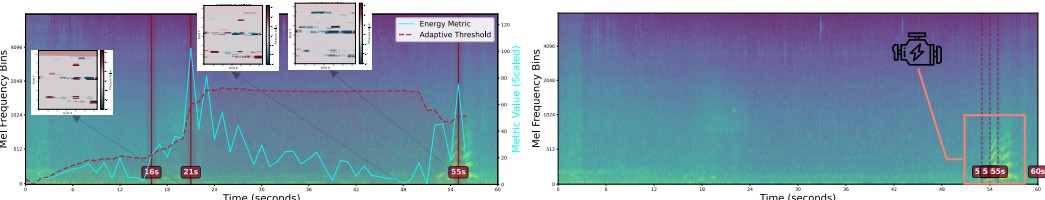

(a) **R0002 (Place d'Armes, Montreal)**: Car engine onset at 53 s; BioWM *p*-field activation near (30, 52).

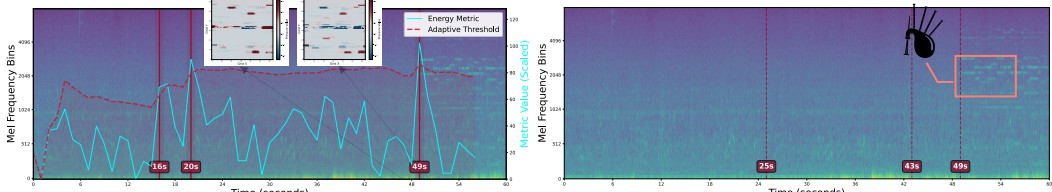

(b) **R0056 (Alexanderplatz, Berlin)**: Bagpipe onset at 49 s; BioWM *p*-field activation near (25, 22).

Figure 2: **BioWM detection of novel events.** Mel-spectrograms with car engine (a) and bagpipe (b) onsets. Left: BioWM outputs (cyan = energy change, red = adaptive threshold, *p*-field states shown); Right: cosine similarity baseline. Vertical dashed lines mark detected drifts.

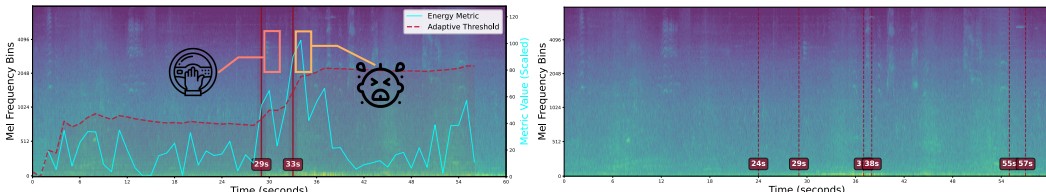

(a) **R0037 (Johnston Road, Hong Kong)**: A baby cry with short pauses. The baseline repeatedly flags pauses, while BioWM registers one event at 33 s.

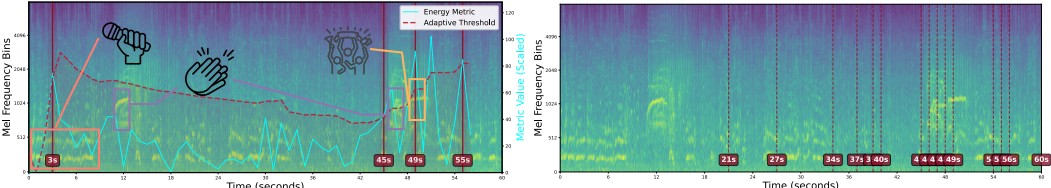

(b) **R0016 (Quincy Market, Boston)**: Festival scene with speech followed by applause. The baseline reports near-continuous changes, while BioWM yields a single detection.

Figure 3: **BioWM is robust to transient pauses.** Mel-spectrograms comparing BioWM outputs (left) and cosine-similarity baseline (right). Vertical dashed lines mark detected drifts; in BioWM plots, cyan = energy change, red = adaptive threshold.

We illustrate this case with two representative recordings in which the most salient novel event occurs near the end of the segment. In both recordings, the background soundscape is relatively stable: the Montreal recording features continuous traffic flow and bird chirping, while the Berlin recording contains crowd conversations, birds, and fountain noise. This provides a clear contrast when a new source emerges late in the sequence, allowing us to assess how well each method captures such onsets.

As shown in Fig. 2, both BioWM and the cosine-similarity baseline detect the late-arriving novel event. However, the baseline often produces spurious drift detections due to its sensitivity to spectral

variance in CNN embeddings (e.g., continuous triggers in Example R0002 (Fig. 2a)). In contrast, BioWM localizes the onset more stably: its $p$-field internal states concentrate energy around the true change point rather than scattering across the segment. This demonstrates BioWM's robustness in distinguishing genuine novel events from background variability.

### 3.3.2 ROBUSTNESS TO TRANSIENT PAUSES

We next consider recordings where the salient sound events include natural pauses or interruptions. Such cases are challenging because a detector may mistake short gaps within an ongoing event for the onset of new events. The first recording (Fig. 3a) was captured in a dense traffic environment, while the second (Fig. 3b) comes from a lively public square during a music festival. In both recordings, prominent sources exhibit intermittent activity, providing a useful testbed for evaluating robustness to transient pauses.

As shown in Fig. 3, both BioWM and the cosine-similarity baseline identify the main events. However, the baseline produces frequent spurious detections whenever short silences or spectral fluctuations occur, leading to repeated triggers in the middle and end portions of Example R0016. In contrast, BioWM consolidates these interruptions into a single detection, demonstrating its ability to maintain stable event representation despite transient pauses.

### 3.3.3 SENSITIVITY TO SUBCATEGORY-LEVEL DRIFT

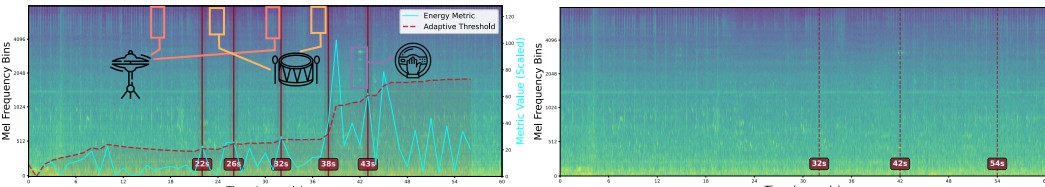

Figure 4: **BioWM sensitivity to subcategory-level drift in Example R0010 (Square Phillips, Montreal — street traffic).** The segment features alternating hi-hat and kick drum patterns, with a distinct car horn at 42 s that BioWM detects as a novel event alongside the subcategory drift. Left: BioWM outputs; Right: cosine similarity baseline. Vertical dashed red lines mark detected drift points. In the left panel, the cyan curve shows energy change, the red curve the adaptive threshold, and detected events are highlighted at the change points.

We further analyze cases where acoustic variation arises within a subcategory of ongoing sounds rather than from the introduction of a completely new source. In this example (Fig. 4), recorded in Square Phillips (Montreal), the background consists of street traffic mixed with music containing two main instruments: a hi-hat at higher frequencies and a slower kick drum. The interplay of these instruments produces several subcategory-level shifts.

Specifically, the hi-hat drops out at 21 s, leaving only the kick drum; it reappears at 32 s and pauses again at 38 s. BioWM successfully detects each of these changes, whereas the cosine-similarity baseline captures only the re-entry at 32 s. At 42 s, a distinct car horn emerges; both methods identify this event as a clear novel source. These results demonstrate BioWM's ability to capture fine-grained subcategory-level drift while still maintaining sensitivity to salient novel events.

### 3.4 SPECTRAL ANALYSIS OF $p$-FIELD DYNAMICS VIA FFT

We computed Fast Fourier Transforms (FFTs) of $p$-field activity, sampling internal states every second to match the sliding window stride (see Subsection 3.1). With a step size of $dt = 0.01$, one second equaled 100 time steps. For each $p$ neuron, we extracted the frequency with maximal FFT amplitude to construct dominant-frequency maps (Fig. 5, Examples R0016 and R0056). To suppress spurious numerical contributions, only neurons above the 75th percentile of variance were retained.

The analysis revealed spatially clustered oscillatory activity rather than uniform grid activation, with dominant frequencies limited to 0–50 Hz by the Nyquist bound ($dt = 0.01$ s). Distinct frequency bands aligned with canonical neural regimes: during stable background periods, both examples ex-

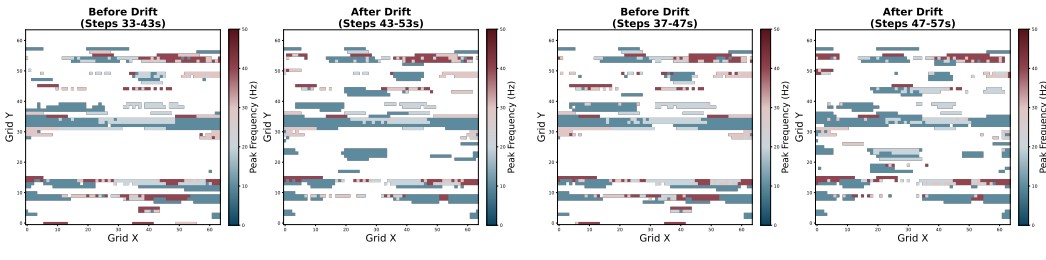

(a) Example R0016 (drift around 43 s)  (b) Example R0056 (drift around 47 s)

Figure 5: **Temporal frequency analysis around drift detection events.** Frequency distributions in active $p$ neurons during 10 s before (left) and after (right) drift onset. Only neurons above the 75th percentile activity threshold are shown.

hibited sustained $\beta$-band activity (15–30 Hz), consistent with its theorized role in working memory maintenance (Lundqvist et al., 2018). After drift onset, Example R0016 shifted toward $\gamma$-band activity (30–50 Hz), reflecting rapid encoding of salient auditory input (applause, cheering). In Example R0056, the post-drift segment with emerging bagpipe sounds showed subsets of neurons entering the $\gamma$ range, while pronounced $\beta$-band oscillations persisted, suggesting mixed maintenance and encoding dynamics. Spatially, both examples exhibited a redistribution of activity: channels along the upper grid boundary (Y=1 row), previously speech-related, showed reduced activity post-drift, whereas deeper clusters ($Y \approx 20$) became strongly engaged, consistent with recruiting new resources for encoding applause, cheering, and musical instruments. These patterns underscore that BioWM reallocates oscillatory dynamics to novel salient sources rather than sustaining prior speech inputs.

In contrast, $\alpha$-band activity (8–12 Hz) remained weak and showed no systematic changes across drift events, precluding confirmation of reported links between elevated $\alpha$ power and attentional lapses (Lakatos et al., 2016; Kasten et al., 2024). Likewise, $\theta$-band activity (4–8 Hz) was sparse and failed to form robust clusters, despite prior reports of selective $\theta$ entrainment supporting auditory working memory (Albouy et al., 2017; Bonetti et al., 2024).

In sum, BioWM drift detection reflects localized, frequency-specific oscillatory clusters rather than uniform grid activation. These band-limited dynamics parallel cortical oscillatory organization while conforming to the propagation and stability constraints of the BioWM framework.

### 3.5 DRIFT DETECTION RATE (DDR)

To complement the qualitative analysis, we report the Drift Detection Rate (DDR), defined as the number of detected drifts divided by the duration of the audio segment. Figure K.1 compares DDR across methods. BioWM consistently yields lower detection activity, with DDR values concentrated around 5%. In contrast, cosine similarity produces a wider and sparser distribution, ranging from 0% to 26% and peaking between 0% and 12%. These results indicate that BioWM effectively detects drift events while avoiding over-sensitivity. The description of the drifts (by both methods) by higher cognition is detailed in Appendix K.

## 4 CONCLUSION

We introduced NAACA, a neuro-inspired framework for online auditory pattern drift detection, with BioWM as its core working-memory component. The contribution of our approach lies in combining a wave-based recurrent field model with an energy-driven drift detection mechanism that adaptively reallocates attention without long-term historical data or offline pretraining. Through theoretical analysis, we proved that binary and striped wave-speed distributions optimize sensitivity to drift, and through experiments on urban soundscapes we demonstrated that BioWM reliably captures novel events, remains robust to transient pauses, and detects subcategory-level shifts more effectively than similarity-based baselines. Furthermore, BioWM exhibits oscillatory dynamics aligned with cortical working memory, underscoring both its biological plausibility and interpretability. Together, these

theoretical and empirical results establish BioWM as a computationally efficient and neuro-inspired foundation for extending long-context reasoning in ALMs and multimodal systems.

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

APPENDIX

# A  RELATED WORK

Auditory selective attention and temporal pattern drift have been studied both in neuroscience, where oscillatory dynamics are linked to memory and attentional control, and in machine learning, where attention bottlenecks in audio language models (ALMs) constrain long-sequence processing. Insights from neuroscience on how the brain selectively tracks relevant inputs and adapts to temporal drift motivate analogous strategies in ALMs, where concept drift detection may provide a principled approach to reallocating limited attention resources.

## A.1  OSCILLATORY DYNAMICS IN MEMORY AND ATTENTION

One prevailing hypothesis is that working memory is supported by discrete neural activity configurations, often described as attractor states (Brennan & Proekt, 2023). Neural oscillations in specific frequency bands have been shown to play a central role in memory, selective attention, and sensitivity to temporal drift. For example, Lundqvist et al. (2018) demonstrated that working memory tasks involve non-stationary dynamics, with gamma bursts during encoding and beta bursts during maintenance. Similarly, selective entrainment of theta oscillations has been shown to enhance auditory working memory performance (Albouy et al., 2017; Bonetti et al., 2024). In contrast, high alpha-band activity has been associated with increased error rates and reduced auditory attention (Lakatos et al., 2016; Kasten et al., 2024).

## A.2  ATTENTION LIMITATIONS AND CONCEPT DRIFT DETECTION

Inspired by these oscillatory mechanisms, one can view attention in ALMs as a resource that must be selectively and dynamically allocated in response to temporal pattern drift rather than distributed uniformly across the input. Although recent advances in ALMs have enabled significant progress in audio understanding, long-form reasoning remains limited by restricted attention span, motivating efforts to extend context length (Wu et al., 2023; He et al., 2024; Bai et al., 2024). Most existing solutions require retraining or fine-tuning, which is computationally costly and inflexible. A complementary perspective is to frame attention allocation as a drift detection problem, where established methods from the machine learning literature may provide efficient and adaptive alternatives.

Given the absence of ground-truth labels in real-world scenarios, we focus on unsupervised approaches. Label-dependent methods, such as PUDD (Lu et al., 2025), fall outside our scope. Cluster-based approaches (Chan et al., 2025) assume sufficient samples per category, which is unrealistic for heterogeneous soundscapes. MCD-DD (Wan et al., 2024) employs contrastive learning on encoder representations, but requires long-term historical data and incurs high computational overhead. The most relevant work, DriftLens (Greco et al., 2025), detects drift from deep representations in real time using both offline and online phases, though its reliance on representation-provider training data limits its adaptability.

# B  DETAILED ALOGORITHMS

This section provides the complete algorithmic specifications for the BioWM-based drift detection framework presented in Subsection 2.1. We present two key algorithms: the main drift detection pipeline and the adaptive threshold computation mechanism.

## B.1  MAIN DRIFT DETECTION ALGORITHM

Algorithm B.1 describes the complete workflow for detecting auditory pattern drift using the NAACA. The algorithm processes streaming audio input through several key stages: feature extraction via a pretrained encoder, oscillatory signal generation, BioWM state updates, energy-based change detection, and persistence filtering.

The energy acceleration (second derivative) within the well-designed structure of the BioWM system captures sudden transitions in the rate of energy change, which correspond to significant shifts in au-

---

**Algorithm B.1** BioWM-based Auditory Pattern Drift Detection

---

**Require:** Audio stream $\{\mathbf{x}_t\}$, encoder $\text{Enc}(\cdot)$, BioWM model with spatial operator $\mathcal{S}$
**Require:** Persistence duration $P = 3$, cooldown period $C = 3$
  1: Initialize energy calculator
  2: Initialize adaptive threshold $T_{\text{adapt}}$ for energy metric
  3: Initialize detection buffer $\mathcal{D} = \emptyset$, cooldown timer $t_{last} = -1$
  4: **for** each time step $t$ **do**
  5:      $\mathbf{p}_t \leftarrow \text{Enc}(\mathbf{x}_t)$            ▷ *Extract event probabilities*
  6:      Generate oscillatory inputs: $S_i(x, t) = a_i(t) \sin(\omega_i t) \mathbf{1}_{\Omega_i}(x)$
  7:      Update BioWM state with oscillatory drive signals
  8:      Calculate current energy acceleration from BioWM dynamics    ▷ *Energy acceleration*
  9:      $T_{\text{adapt}} \leftarrow \mu + 2\sigma(1 + \alpha \cdot \text{trend})$      ▷ *Update adaptive threshold*
10:      $d_{candidate} \leftarrow \mathbf{1}(\text{energy acceleration} > T_{\text{adapt}})$      ▷ *Energy-based detection*
11:      Add $d_{candidate}$ to detection buffer $\mathcal{D}$      ▷ *Persistence filtering*
12:      **if** $|\mathcal{D}| \geq P$ **then**
13:          $r_{persist} \leftarrow \frac{1}{P} \sum_{i=0}^{P-1} \mathcal{D}[-P + i]$      ▷ *Persistence ratio*
14:          **if** $r_{persist} \geq 0.5$ AND $t - t_{last} > C$ **then**
15:              **output** Drift detected at time $t$
16:              $t_{last} \leftarrow t$, clear $\mathcal{D}$      ▷ *Reset detection state*
17:          **end if**
18:      **end if**
19: **end for**

---

**Algorithm B.2** Online Adaptive Threshold Calculation

---

**Require:** Window size $W = 20$, trend adjustment factor $\alpha = 0.2$
  1: Initialize value buffer $\mathcal{V} = \emptyset$ with maximum size $W$
  2: **function** AdaptiveThreshold.update($v_{new}$)
  3: Add $v_{new}$ to buffer $\mathcal{V}$
  4: **if** $|\mathcal{V}| < 5$ **then**
  5:      $\mu \leftarrow \frac{1}{|\mathcal{V}|} \sum_i \mathcal{V}_i$      ▷ *Insufficient data for robust statistics*
  6:      **if** $|\mathcal{V}| > 1$ **then**
  7:          $\sigma \leftarrow \sqrt{\frac{1}{|\mathcal{V}|-1} \sum_i (\mathcal{V}_i - \mu)^2}$
  8:      **else**
  9:          $\sigma \leftarrow 0.1$
10:      **end if**
11:      **return** $\mu + 1.5\sigma$      ▷ *Simple threshold for bootstrap*
12: **end if**      ▷ *Compute baseline statistics*
13: $\mu \leftarrow \frac{1}{|\mathcal{V}|} \sum_i \mathcal{V}_i$      ▷ *Running mean*
14: $\sigma \leftarrow \sqrt{\frac{1}{|\mathcal{V}|-1} \sum_i (\mathcal{V}_i - \mu)^2}$      ▷ *Running std*
15: **if** $|\mathcal{V}| \geq 3$ **then**
16:      $\mathbf{x} \leftarrow [0, 1, \ldots, |\mathcal{V}| - 1]$      ▷ *Time indices*
17:      slope $\leftarrow \frac{\sum_i (x_i - \bar{x})(\mathcal{V}_i - \mu)}{\sum_i (x_i - \bar{x})^2}$      ▷ *Linear regression slope*
18:      $f_{trend} \leftarrow \frac{|\text{slope}|}{\sigma + 10^{-8}}$      ▷ *Normalized trend strength*
19: **else**
20:      $f_{trend} \leftarrow 0$
21: **end if**
22: $T_{adapt} \leftarrow \mu + 2\sigma(1 + \alpha \cdot f_{trend})$      ▷ *Adaptive threshold with trend adjustment*
23: **return** $T_{adapt}$

---

ditory patterns. The algorithm employs an adaptive threshold mechanism that automatically adjusts to the system's baseline behavior and temporal trends, eliminating the need for manual threshold tuning.

To ensure robust detection and minimize false positives, the algorithm incorporates persistence filtering that requires a minimum proportion of recent detections before confirming a drift event. Additionally, a cooldown period prevents redundant detections of the same drift event.

### B.2 Adaptive Threshold Computation

Algorithm B.2 details the online computation of adaptive thresholds for energy-based drift detection. The threshold adapts to both the statistical properties of recent observations and temporal trends in the data.

The algorithm maintains a sliding window of recent metric values and computes the threshold as $T = \mu + 2\sigma(1 + \alpha \cdot f_{\text{trend}})$, where $\mu$ and $\sigma$ are the running mean and standard deviation, $\alpha$ is the trend adjustment factor, and $f_{\text{trend}}$ quantifies the strength of temporal trends using linear regression.

During the initial bootstrap phase with insufficient data (fewer than 5 samples), the algorithm uses a simplified threshold computation to avoid instability. The trend factor captures whether the metric values are systematically increasing or decreasing, allowing the threshold to adapt accordingly. This prevents false negatives during periods of natural system evolution while maintaining sensitivity to abrupt changes.

The adaptive nature of this threshold computation is crucial for handling diverse auditory environments with varying baseline activity levels, ensuring consistent detection performance across different acoustic contexts without requiring environment-specific calibration.

## C  Proof for Theorem 1

*Proof.* The BioWM system is governed by the discrete formulation from Eqs. 4.

For a local grid point $(x, y)$, define the state vector $\mathbf{h} = [p, v_x, v_y]^T$. The system becomes:

$$\mathbf{h}(t + \Delta t) = \mathbf{A}\mathbf{h}(t) + S(x, y, t) \tag{C.1}$$

where the system matrix is:

$$\mathbf{A} = \mathbf{M}_{\text{Damp}}^{-1}\mathbf{M}_{\text{Velocity}} \tag{C.2}$$

$$\mathbf{M}_{\text{Velocity}} = \begin{bmatrix} 1 & -c^2\Delta t\frac{\partial}{\partial x} & -c^2\Delta t\frac{\partial}{\partial y} \\ -\Delta t\frac{\partial}{\partial x} & 1 & 0 \\ -\Delta t\frac{\partial}{\partial y} & 0 & 1 \end{bmatrix} \tag{C.3}$$

$$\mathbf{M}_{\text{Damp}} = \begin{bmatrix} 1 + \Delta t k^p(x, y) & 0 & 0 \\ 0 & 1 + \Delta t k^v(x, y) & 0 \\ 0 & 0 & 1 + \Delta t k^v(x, y) \end{bmatrix} \tag{C.4}$$

In Fourier space, spatial derivatives become algebraic operations:

$$\frac{\partial}{\partial x} \to i\xi_x \tag{C.5}$$

$$\frac{\partial}{\partial y} \to i\xi_y \tag{C.6}$$

$$\nabla \cdot \mathbf{v} \to i(\xi_x v_x + \xi_y v_y) \tag{C.7}$$

The system matrix in Fourier domain becomes:

$$\mathbf{A} = \mathbf{M}_{\text{Damp}}^{-1}\mathbf{M}_{\text{Velocity}} = \begin{bmatrix} \frac{1}{1+\Delta t k_{i,j}^p} & -\frac{c_{i,j}^2\Delta t i\xi_x}{1+\Delta t k_{i,j}^p} & -\frac{c_{i,j}^2\Delta t i\xi_y}{1+\Delta t k_{i,j}^p} \\ -\frac{\Delta t i\xi_x}{1+\Delta t k_{i,j}^o} & \frac{1}{1+\Delta t k_{i,j}^o} & 0 \\ -\frac{\Delta t i\xi_y}{1+\Delta t k_{i,j}^o} & 0 & \frac{1}{1+\Delta t k_{i,j}^o} \end{bmatrix} \tag{C.8}$$

The characteristic polynomial $\det(\mathbf{A} - \lambda\mathbf{I}) = 0$ expands to:

$$\left(\frac{1}{1 + \Delta t k_{i,j}^o} - \lambda\right)^2 \left[\left(\frac{1}{1 + \Delta t k_{i,j}^p} - \lambda\right)\left(\frac{1}{1 + \Delta t k_{i,j}^o} - \lambda\right) + \frac{c_{i,j}^2 \Delta t^2 (\xi_x^2 + \xi_y^2)}{(1 + \Delta t k_{i,j}^p)(1 + \Delta t k_{i,j}^o)}\right] = 0 \quad \text{(C.9)}$$

This yields one real eigenvalue:

$$\lambda_1 = \frac{1}{1 + \Delta t k_{i,j}^o} \quad \text{(C.10)}$$

The other two eigenvalues satisfy:

$$\left(\frac{1}{1 + \Delta t k_{i,j}^p} - \lambda\right)\left(\frac{1}{1 + \Delta t k_{i,j}^o} - \lambda\right) + \frac{c_{i,j}^2 \Delta t^2 (\xi_x^2 + \xi_y^2)}{(1 + \Delta t k_{i,j}^p)(1 + \Delta t k_{i,j}^o)} = 0 \quad \text{(C.11)}$$

Solving the quadratic equation:

$$\lambda_{2,3} = \frac{1}{2}\left[\frac{1}{1 + \Delta t k_{i,j}^p} + \frac{1}{1 + \Delta t k_{i,j}^o}\right] \pm \sqrt{\Delta^2} \quad \text{(C.12)}$$

where the discriminant is:

$$\Delta^2 = \left(\frac{1}{1 + \Delta t k_{i,j}^p} - \frac{1}{1 + \Delta t k_{i,j}^o}\right)^2 - \frac{4c_{i,j}^2 \Delta t^2 (\xi_x^2 + \xi_y^2)}{(1 + \Delta t k_{i,j}^p)(1 + \Delta t k_{i,j}^o)} \quad \text{(C.13)}$$

For typical parameter ranges where the discriminant is negative, we obtain complex conjugate pairs:

$$\lambda_{2,3} = \frac{1}{2}\left[\frac{1}{1 + \Delta t k_{i,j}^p} + \frac{1}{1 + \Delta t k_{i,j}^o}\right] \pm i\frac{c_{i,j}\Delta t\sqrt{\xi_x^2 + \xi_y^2}}{\sqrt{(1 + \Delta t k_{i,j}^p)(1 + \Delta t k_{i,j}^o)}} \quad \text{(C.14)}$$

The phase angle of the complex eigenvalue is:

$$\theta = \tan^{-1}\left(\frac{\mathrm{Im}(\lambda)}{\mathrm{Re}(\lambda)}\right) = \tan^{-1}\left(\frac{2c_{i,j}\Delta t\sqrt{\xi_x^2 + \xi_y^2}}{\sqrt{(1 + \Delta t k_{i,j}^p)(1 + \Delta t k_{i,j}^o)}\left(\frac{1}{1 + \Delta t k_{i,j}^p} + \frac{1}{1 + \Delta t k_{i,j}^o}\right)}\right) \quad \text{(C.15)}$$

In discrete-time systems, the frequency corresponding to a complex eigenvalue with phase angle $\theta$ is:

$$f = \frac{\theta}{\pi\Delta t} \quad \text{(C.16)}$$

Therefore:

$$f_{i,j} = \frac{1}{\pi\Delta t}\tan^{-1}\left(\frac{2c_{i,j}\Delta t\sqrt{\xi_x^2 + \xi_y^2}}{\sqrt{(1 + \Delta t k_{i,j}^o)(1 + \Delta t k_{i,j}^o)}\left(\frac{1}{1 + \Delta t k_{i,j}^p} + \frac{1}{1 + \Delta t k_{i,j}^o}\right)}\right) \quad \text{(C.17)}$$

This completes the derivation. $\qquad\square$

## D  PROOF FOR THEOREM 2

*Proof.* We prove the equivalence by transforming the system step by step. We begin by differentiating the pressure equation from Eqs. 3 with respect to time:

$$\frac{\partial^2 p}{\partial t^2} + k^p\frac{\partial p}{\partial t} = -c^2\left(\frac{\partial^2 v_x}{\partial x\partial t} + \frac{\partial^2 v_y}{\partial y\partial t}\right) + \frac{\partial S}{\partial t}. \quad \text{(D.1)}$$

To evaluate the velocity derivatives, we use the velocity equations from Eqs. 3:

$$\frac{\partial v_x}{\partial t} = -k^v v_x - \frac{\partial p}{\partial x}, \tag{D.2}$$

$$\frac{\partial v_y}{\partial t} = -k^v v_y - \frac{\partial p}{\partial y}. \tag{D.3}$$

Differentiating Eqs. D.2 and D.3 spatially, we obtain:

$$\frac{\partial^2 v_x}{\partial x \partial t} = -k^v \frac{\partial v_x}{\partial x} - \frac{\partial^2 p}{\partial x^2}, \tag{D.4}$$

$$\frac{\partial^2 v_y}{\partial y \partial t} = -k^v \frac{\partial v_y}{\partial y} - \frac{\partial^2 p}{\partial y^2}. \tag{D.5}$$

Adding Eqs. D.4 and D.5 together yields:

$$\frac{\partial^2 v_x}{\partial x \partial t} + \frac{\partial^2 v_y}{\partial y \partial t} = -k^v \left( \frac{\partial v_x}{\partial x} + \frac{\partial v_y}{\partial y} \right) - \nabla^2 p. \tag{D.6}$$

We now substitute the divergence relation from Eqs. 3:

$$\frac{\partial v_x}{\partial x} + \frac{\partial v_y}{\partial y} = -\frac{1}{c^2} \left( \frac{\partial p}{\partial t} + k^p p - S \right). \tag{D.7}$$

Inserting Eq. D.7 into Eq. D.6 gives us:

$$\frac{\partial^2 v_x}{\partial x \partial t} + \frac{\partial^2 v_y}{\partial y \partial t} = \frac{k^v}{c^2} \left( \frac{\partial p}{\partial t} + k^p p - S \right) - \nabla^2 p. \tag{D.8}$$

Substituting Eq. D.8 back into Eq. D.1:

$$\frac{\partial^2 p}{\partial t^2} + k^p \frac{\partial p}{\partial t} = -c^2 \left[ \frac{k^v}{c^2} \left( \frac{\partial p}{\partial t} + k^p p - S \right) - \nabla^2 p \right] + \frac{\partial S}{\partial t}. \tag{D.9}$$

Simplifying Eq. D.9, we get:

$$\frac{\partial^2 p}{\partial t^2} + k^p \frac{\partial p}{\partial t} = -k^v \left( \frac{\partial p}{\partial t} + k^p p - S \right) + c^2 \nabla^2 p + \frac{\partial S}{\partial t}. \tag{D.10}$$

Finally, rearranging terms in Eq. D.10 leads to:

$$\frac{\partial^2 p}{\partial t^2} + (k^p + k^v)\frac{\partial p}{\partial t} + k^v k^p p = c^2 \nabla^2 p + \left( k^v + \frac{\partial}{\partial t} \right) S, \tag{D.11}$$

which matches Eq. 8. Thus, the system Eqs. 3 is equivalent to the second-order damped wave equation with the stated coefficients. □

# E  PROOF FOR THEOREM 3

*Proof.* We begin by computing the time derivative of energy:

$$\frac{dE}{dt} = \iint \left[ p\frac{\partial p}{\partial t} + v_x \frac{\partial v_x}{\partial t} + v_y \frac{\partial v_y}{\partial t} \right] dx\, dy. \tag{E.1}$$

Substituting the system equations from Eqs. 3 into Eq. E.1, we obtain:

$$\frac{dE}{dt} = \iint \left[ p\big(-k^p p - c^2 (\tfrac{\partial v_x}{\partial x} + \tfrac{\partial v_y}{\partial y}) + S\big) + v_x\big(-k^v v_x - \tfrac{\partial p}{\partial x}\big) + v_y\big(-k^v v_y - \tfrac{\partial p}{\partial y}\big) \right] dx\, dy. \tag{E.2}$$

Expanding the terms in Eq. E.2 yields:

$$\frac{dE}{dt} = \iint \left[ -k^p p^2 - c^2 p\left(\frac{\partial v_x}{\partial x} + \frac{\partial v_y}{\partial y}\right) + pS - k^v v_x^2 - v_x \frac{\partial p}{\partial x} - k^v v_y^2 - v_y \frac{\partial p}{\partial y} \right] dx\, dy. \quad \text{(E.3)}$$

We can group the terms in Eq. E.3 to separate dissipation, coupling, and source contributions:

$$\frac{dE}{dt} = \iint \left[ -k^p p^2 - k^v (v_x^2 + v_y^2) \right] dx\, dy + \iint \left[ -c^2 p\left(\frac{\partial v_x}{\partial x} + \frac{\partial v_y}{\partial y}\right) - v_x \frac{\partial p}{\partial x} - v_y \frac{\partial p}{\partial y} \right] dx\, dy + \iint pS\, dx\, dy. \quad \text{(E.4)}$$

To handle the coupling terms in Eq. E.4, we apply integration by parts. For example:

$$\iint v_x \frac{\partial p}{\partial x} dx\, dy = \left. v_x p \right|_{x \text{ boundaries}} - \iint p \frac{\partial v_x}{\partial x} dx\, dy. \quad \text{(E.5)}$$

Since periodic boundary conditions imply the boundary term vanishes, we have:

$$\iint v_x \frac{\partial p}{\partial x} dx\, dy = - \iint p \frac{\partial v_x}{\partial x} dx\, dy. \quad \text{(E.6)}$$

Similarly, for the $y$-component:

$$\iint v_y \frac{\partial p}{\partial y} dx\, dy = - \iint p \frac{\partial v_y}{\partial y} dx\, dy. \quad \text{(E.7)}$$

Using Eqs. E.6 and E.7, the coupling terms in Eq. E.4 reduce to:

$$-c^2 p \left( \frac{\partial v_x}{\partial x} + \frac{\partial v_y}{\partial y} \right) - v_x \frac{\partial p}{\partial x} - v_y \frac{\partial p}{\partial y}$$

$$= -c^2 p \left( \frac{\partial v_x}{\partial x} + \frac{\partial v_y}{\partial y} \right) + p \left( \frac{\partial v_x}{\partial x} + \frac{\partial v_y}{\partial y} \right) \quad \text{(E.8)}$$

$$= -(c^2 - 1)\, p \left( \frac{\partial v_x}{\partial x} + \frac{\partial v_y}{\partial y} \right).$$

Combining all results, we obtain the final expression:

$$\frac{dE}{dt} = - \iint \left[ k^p p^2 + k^v (v_x^2 + v_y^2) \right] dx\, dy - \iint (c^2 - 1) p \left( \frac{\partial v_x}{\partial x} + \frac{\partial v_y}{\partial y} \right) dx\, dy + \iint pS\, dx\, dy, \quad \text{(E.9)}$$

which establishes the claim. $\qquad \square$

# F PROOF FOR THEOREM 4

## F.1 STATEMENTS AND PROOF OF LEMMAS FOR THEOREM 4

**Lemma 1** (Interface Reflection Analysis). *Consider a plane wave incident on an interface between two constant-speed media with speeds $c_1$ and $c_2$. With uniform density $\rho = 1$ so that impedance $Z = \rho c = c$, the pressure reflection coefficient is*

$$R = \frac{c_2 - c_1}{c_1 + c_2}. \quad \text{(F.1)}$$

*The energy reflection ratio is*

$$\frac{E_{\text{reflected}}}{E_{\text{incident}}} = |R|^2. \quad \text{(F.2)}$$

*Under a fixed speed-sum constraint $c_1 + c_2 = C$, $|R|^2$ is maximized by extreme (binary) contrasts $(c_1, c_2) \in \{(0, C), (C, 0)\}$, while $c_1 = c_2 = C/2$ yields $R = 0$.*

*Proof.* With $\rho = 1$, the medium impedances are $Z_i = c_i$, $i = 1, 2$. Continuity of pressure and normal velocity (using $v = p/Z$) at the interface yields

$$p_1 + p_1^r = p_2^t, \tag{F.3}$$

$$\frac{p_1}{Z_1} - \frac{p_1^r}{Z_1} = \frac{p_2^t}{Z_2}, \tag{F.4}$$

where subscripts denote incident (no subscript), reflected ($r$), and transmitted ($t$) components. Solving these equations gives

$$p_2^t = \frac{2Z_2}{Z_1 + Z_2}\, p_1, \tag{F.5}$$

$$\frac{p_1^r}{p_1} = \frac{Z_2 - Z_1}{Z_1 + Z_2} = \frac{c_2 - c_1}{c_1 + c_2} = R. \tag{F.6}$$

The reflected-to-incident energy ratio is therefore

$$\frac{E_{\text{reflected}}}{E_{\text{incident}}} = |R|^2 = \left(\frac{c_2 - c_1}{c_1 + c_2}\right)^2. \tag{F.7}$$

If $c_1 + c_2 = C$, maximizing $|R|^2 = \left(\frac{c_2 - c_1}{C}\right)^2$ is equivalent to maximizing $c_2 - c_1$. The maximum occurs at the binary extremes $(c_1, c_2) = (0, C)$ or $(C, 0)$, for which $|R| \to 1$. Any interior choice yields a smaller $R$. For example, $c_1 = c_2 = C/2$ gives $R = 0$, while $c_1 = 0.9C$, $c_2 = 0.1C$ gives $|R| = 0.8 < 1$. $\qquad\square$

**Lemma 2** (Interface Density Maximization). *For a discrete field $c(i,j)$ on a periodic 2D grid, define the total variation (contrast budget)*

$$\text{TV}[c] = \sum_{i,j} \Big( |c(i+1, j) - c(i,j)| + |c(i, j+1) - c(i,j)| \Big). \tag{F.8}$$

*With local per-interface transfer efficiency $f(\Delta c) \propto (\Delta c)^2$ (motivated by $|R|^2 \propto (\Delta c)^2$ when $c_1 + c_2$ is approximately constant), and a fixed budget*

$$\sum_k \Delta c_k \leq B, \qquad \Delta c_k \geq 0, \tag{F.9}$$

*the total efficiency $\sum_k f(\Delta c_k)$ is maximized by concentrating contrast on as few interfaces as possible (binary extreme allocations), dominating uniform spread.*

*Proof.* Let $f(\Delta c) \propto (\Delta c)^2$. The objective is to maximize $\sum_{k=1}^N (\Delta c_k)^2$ subject to Eq. F.9.

For a uniform allocation,

$$\Delta c_k = \tfrac{B}{N} \ \forall k \implies \sum_{k=1}^N (\Delta c_k)^2 = \tfrac{B^2}{N}. \tag{F.10}$$

For a binary allocation,

$$(\Delta c_1, \ldots, \Delta c_N) = (B, 0, \ldots, 0) \implies \sum_{k=1}^N (\Delta c_k)^2 = B^2. \tag{F.11}$$

Since $B^2 > B^2/N$ for all $N > 1$, binary allocation yields a strictly larger value. Moreover, as the function $x \mapsto x^2$ is convex on $x \geq 0$, the maximum of $\sum_k (\Delta c_k)^2$ under the budget constraint is attained at an extreme point, i.e., a concentrated (binary) allocation. Hence, with a fixed TV budget, binary distributions maximize interface contrast density and transfer. $\qquad\square$

**Lemma 3** (Change Detection Sensitivity). *Define instantaneous energy-rate sensitivity to source perturbations $\delta S$ by*

$$\delta\left(\frac{dE}{dt}\right) = \iint p(x, y, t)\, \delta S(x, y, t)\, dx\, dy. \tag{F.12}$$

*Define the change detection sensitivity*

$$\Psi = \mathbb{E}\left[\left\|\iint p(x,y,t)\,\delta S(x,y,t)\,dx\,dy\right\|\right]. \tag{F.13}$$

*Binary contrasts maximize $\Psi$ by amplifying $p$ through three complementary mechanisms: strong interface reflections ($|R| \to 1$), redistribution-driven concentration across high-c/low-c regions, and enhanced multi-mode coupling and coherence.*

*Proof.* We begin by introducing the sensitivity measure,

$$\delta\left(\frac{dE}{dt}\right) = \iint p\,\delta S\,dx\,dy, \qquad \Psi = \mathbb{E}\left[\left\|\iint p\,\delta S\,dx\,dy\right\|\right], \tag{F.14}$$

where $\mathbb{E}[\cdot]$ denotes the expected value over time and input variations. It follows that maximizing $\Psi$ requires amplification of $|p|$ in regions where $\delta S$ varies.

In order to account for this amplification, several mechanisms must be considered. First, interface amplification (from Lemma 1) drives $|R| \to 1$, thereby producing strong standing waves with

$$|p|_{\max} \approx |p_{\text{incident}}| + |p_{\text{reflected}}| \approx 2|p_{\text{incident}}|. \tag{F.15}$$

Second, spatial variation in $c$ induces a redistribution of energy between velocity-like and pressure-like components, which enhances local values of $p$. Finally, modal coupling provides an additional contribution: stronger contrast increases coupling strengths and enables coherent multi-mode amplification.

To consolidate these contributions, we define the aggregated amplification factor as

$$A[c] = \frac{\mathbb{E}\left[|p|_{\text{binary}}^2\right]}{\mathbb{E}[|p|_{\text{smooth}}^2]} = A_{\text{reflect}} \times A_{\text{redistribute}} \times A_{\text{coupling}} \gg 1. \tag{F.16}$$

As a direct consequence, one obtains

$$\Psi[c_{\text{binary}}] = A[c_{\text{binary}}] \times \Psi[c_{\text{smooth}}] \gg \Psi[c_{\text{smooth}}]. \tag{F.17}$$

Therefore, binary contrasts are seen to maximize change detection sensitivity. □

# G   PROOF FOR THEOREM 5

## G.1   STATEMENTS AND PROOF OF LEMMAS FOR THEOREM 5

**Lemma 4** (Modal Coupling Derivation and Selection Rule). *In the periodic domain $[0, L_x] \times [0, L_y]$ with orthonormal Fourier basis*

$$\phi_{m,n}(x,y) = \frac{1}{\sqrt{A}}\exp\left(i\frac{2\pi mx}{L_x} + i\frac{2\pi ny}{L_y}\right), \quad A = L_xL_y, \tag{G.1}$$

*and wavenumbers*

$$k_{m,n}^2 = \left(\frac{2\pi m}{L_x}\right)^2 + \left(\frac{2\pi n}{L_y}\right)^2, \tag{G.2}$$

*write $p(x,y,t) = \sum_{m,n} a_{m,n}(t)\phi_{m,n}(x,y)$ and $c^2(x,y) = c_0^2 + \delta c^2(x,y)$. Here, $a_{m,n}(t)$ are the time-dependent modal coefficients. Neglecting damping, the modal system*

$$\ddot{a}_{m,n} + \omega_{m,n}^2 a_{m,n} = \sum_{m',n'} C_{(m,n),(m',n')} a_{m',n'} + S_{m,n}, \tag{G.3}$$

*has $\omega_{m,n}^2 = c_0^2 k_{m,n}^2$ and coupling*

$$C_{(m,n),(m',n')} = k_{m',n'}^2 \langle \delta c^2, \phi_{m',n'}, \phi_{m,n}\rangle. \tag{G.4}$$

*For striped $\delta c^2(x,y) = f(y)$,*

$$\langle \delta c^2, \phi_{m',n'}, \phi_{m,n}\rangle = \delta_{m,m'}\,\hat{V}_{n-n'}, \tag{G.5}$$

*i.e., coupling is block-diagonal in $m$ and depends on the y-Fourier coefficients $\hat{V}_q$ of $f$, where*

$$\hat{V}_q = \frac{1}{L_y}\int_0^{L_y} f(y)\,e^{i\frac{2\pi qy}{L_y}}\,dy. \tag{G.6}$$

*Proof.* We begin by establishing the modal expansion and deriving the coupling matrix through systematic projection onto Fourier basis functions.

On the periodic domain $[0, L_x] \times [0, L_y]$, we employ the orthonormal Fourier basis:

$$\phi_{m,n}(x,y) = \frac{1}{\sqrt{A}} \exp\left[ i \left( \frac{2\pi m}{L_x} x + \frac{2\pi n}{L_y} y \right) \right], \quad k_{m,n}^2 = \left( \frac{2\pi m}{L_x} \right)^2 + \left( \frac{2\pi n}{L_y} \right)^2, \qquad \text{(G.7)}$$

where $A = L_x L_y$ is the domain area. These functions satisfy $\nabla^2 \phi_{m,n} = -k_{m,n}^2 \phi_{m,n}$.

Starting from the wave equation without damping:

$$\frac{\partial^2 p}{\partial t^2} = c^2(x,y) \nabla^2 p + S(x,y,t), \quad c^2(x,y) = c_0^2 + \delta c^2(x,y), \qquad \text{(G.8)}$$

we expand $p(x,y,t) = \sum_{m,n} a_{m,n}(t) \phi_{m,n}(x,y)$ to obtain:

$$\sum_{m,n} \ddot{a}_{m,n}(t) \phi_{m,n}(x,y) = (c_0^2 + \delta c^2(x,y)) \nabla^2 \left( \sum_{m,n} a_{m,n}(t) \phi_{m,n}(x,y) \right) + S(x,y,t)$$

$$= \sum_{m,n} a_{m,n}(t)(c_0^2 + \delta c^2(x,y)) \nabla^2 \phi_{m,n}(x,y) + S(x,y,t). \qquad \text{(G.9)}$$

Substituting $\nabla^2 \phi_{m,n} = -k_{m,n}^2 \phi_{m,n}$ into Eq. G.9 yields:

$$\sum_{m,n} \ddot{a}_{m,n}(t) \phi_{m,n} = -\sum_{m,n} a_{m,n}(t) k_{m,n}^2 (c_0^2 + \delta c^2) \phi_{m,n} + S(x,y,t). \qquad \text{(G.10)}$$

To project this equation onto individual modes, we define the normalized inner product on $[0, L_x] \times [0, L_y]$ by:

$$\langle f, g \rangle := \frac{1}{A} \int_0^{L_x} \int_0^{L_y} f(x,y) \, g^*(x,y) \, dx \, dy, \qquad A = L_x L_y, \qquad \text{(G.11)}$$

so that the Fourier modes are orthonormal: $\langle \phi_{m,n}, \phi_{\mu,\nu} \rangle = \delta_{m,\mu} \delta_{n,\nu}$, where $\delta$ denotes the Kronecker delta function, which is defined as

$$\delta_{m,\mu} = \begin{cases} 1 & \text{if } m = \mu \\ 0 & \text{if } m \neq \mu \end{cases} \quad \text{and} \quad \delta_{n,\nu} = \begin{cases} 1 & \text{if } n = \nu \\ 0 & \text{if } n \neq \nu \end{cases}.$$

We also use the trilinear shorthand:

$$\langle \delta c^2, \phi_{m,n}, \phi_{\mu,\nu} \rangle := \frac{1}{A} \int_0^{L_x} \int_0^{L_y} \delta c^2(x,y) \, \phi_{m,n}^*(x,y) \, \phi_{\mu,\nu}(x,y) \, dx \, dy. \qquad \text{(G.12)}$$

Multiplying Eq. G.10 by $\phi_{\mu,\nu}(x,y)$ and integrating over the domain, we analyze each term separately. For the left-hand side:

$$\left\langle \sum_{m,n} \ddot{a}_{m,n} \phi_{m,n}, \, \phi_{\mu,\nu} \right\rangle = \sum_{m,n} \ddot{a}_{m,n} \langle \phi_{m,n}, \phi_{\mu,\nu} \rangle = \sum_{m,n} \ddot{a}_{m,n} \delta_{m,\mu} \delta_{n,\nu} = \ddot{a}_{\mu,\nu}(t). \qquad \text{(G.13)}$$

For the reference-speed part:

$$-\left\langle \sum_{m,n} a_{m,n} k_{m,n}^2 c_0^2 \phi_{m,n}, \, \phi_{\mu,\nu} \right\rangle = -c_0^2 \sum_{m,n} a_{m,n} k_{m,n}^2 \langle \phi_{m,n}, \phi_{\mu,\nu} \rangle \qquad \text{(G.14)}$$

$$= -c_0^2 k_{\mu,\nu}^2 a_{\mu,\nu}(t). \qquad \text{(G.15)}$$

For the perturbation (coupling) part:

$$-\left\langle \sum_{m,n} a_{m,n} k_{m,n}^2 \delta c^2 \phi_{m,n}, \, \phi_{\mu,\nu} \right\rangle = -\sum_{m,n} a_{m,n} k_{m,n}^2 \langle \delta c^2, \phi_{m,n}, \phi_{\mu,\nu} \rangle. \qquad \text{(G.16)}$$

For the source part:

$$\langle S(x,y,t), \phi_{\mu,\nu} \rangle =: S_{\mu,\nu}(t). \tag{G.17}$$

Collecting terms from Eqs. G.13-G.17 gives:

$$\ddot{a}_{\mu,\nu}(t) + c_0^2 k_{\mu,\nu}^2 \, a_{\mu,\nu}(t) = -\sum_{m,n} k_{m,n}^2 \langle \delta c^2, \phi_{m,n}, \phi_{\mu,\nu} \rangle \, a_{m,n}(t) + S_{\mu,\nu}(t). \tag{G.18}$$

Defining $\omega_{m,n}^2 := c_0^2 k_{m,n}^2$ and the coupling matrix:

$$C_{(\mu,\nu),(m,n)} := k_{m,n}^2 \langle \delta c^2, \phi_{m,n}, \phi_{\mu,\nu} \rangle, \tag{G.19}$$

and relabeling $(\mu,\nu) \to (m,n)$, $(m,n) \to (m',n')$ in the sum, we obtain the modal system:

$$\ddot{a}_{m,n} + \omega_{m,n}^2 a_{m,n} = \sum_{m',n'} C_{(m,n),(m',n')} \, a_{m',n'} + S_{m,n}. \tag{G.20}$$

For the special case of striped distributions where $\delta c^2(x,y) = f(y)$, we can evaluate the coupling matrix explicitly using Eq. G.12:

$$\langle \delta c^2, \phi_{m',n'}, \phi_{m,n} \rangle = \frac{1}{A} \iint f(y) \, e^{i\frac{2\pi(m-m')x}{L_x}} \, e^{i\frac{2\pi(n-n')y}{L_y}} \, dx \, dy \tag{G.21}$$

$$= \frac{1}{A} \left[ \int_0^{L_x} e^{i\frac{2\pi(m-m')x}{L_x}} dx \right] \left[ \int_0^{L_y} f(y) e^{i\frac{2\pi(n-n')y}{L_y}} dy \right]. \tag{G.22}$$

The $x$-integral in Eq. G.22 evaluates to $L_x \delta_{m,m'}$. Defining the Fourier coefficients:

$$\hat{V}_q = \frac{1}{L_y} \int_0^{L_y} f(y) e^{i\frac{2\pi q y}{L_y}} dy, \quad q = n - n', \tag{G.23}$$

the $y$-integral becomes $L_y \hat{V}_{n-n'}$. Therefore:

$$\langle \delta c^2, \phi_{m',n'}, \phi_{m,n} \rangle = \delta_{m,m'} \hat{V}_{n-n'}. \tag{G.24}$$

Substituting Eq. G.24 into Eq. G.19, the coupling matrix under striped distributions becomes block-diagonal in $m$:

$$C_{(m,n),(m',n')} = k_{m',n'}^2 \delta_{m,m'} \hat{V}_{n-n'}. \tag{G.25}$$

To analyze the total coupling strength, we compute the Frobenius norm of the coupling matrix from Eq. G.25:

$$\|C\|_F^2 = \sum_m \sum_{n,n'} k_{m,n'}^4 \, |\hat{V}_{n'-n}|^2. \tag{G.26}$$

Using Parseval's theorem, $\|C\|_F^2$ is proportional to $\int_0^{L_y} |\delta c^2(y)|^2 dy$, which is maximized by binary-valued contrasts under amplitude constraints.

For specified mode-pair coupling between modes $(m,n)$ and $(m, n+q_0)$, the coupling strength from Eq. G.25 is:

$$|C_{(m,n),(m,n+q_0)}| = k_{m,n+q_0}^2 |\hat{V}_{q_0}|. \tag{G.27}$$

Maximizing $|\hat{V}_{q_0}|$ under mean and amplitude constraints again yields a binary striped $\delta c^2(y)$ with stripe period tuned to $q_0$. $\qquad\square$

**Lemma 5** (Total Coupling Strength Maximization). *For striped perturbations $\delta c^2(x,y) = f(y)$, the Frobenius norm of the coupling matrix is*

$$\|C\|_F^2 = \sum_{m,n} \sum_{m',n'} |C_{(m,n),(m',n')}|^2 = \sum_m \sum_{n,n'} k_{m,n'}^4 \, |\hat{V}_{n-n'}|^2, \tag{G.28}$$

*where $\hat{V}_q$ are the Fourier coefficients of $f(y)$. Using Parseval's identity, this norm is proportional to the squared $L^2$-norm of $\delta c^2(y)$. Under fixed mean and amplitude bounds $c_{\min}^2 \leq c^2 \leq c_{\max}^2$, this is maximized by binary two-level functions, so binary striping maximizes total coupling strength.*

*Proof.* We begin by applying the selection rule from Lemma 4, which shows that striped patterns lead to:

$$C_{(m,n),(m',n')} = k^2_{m',n'} \, \delta_{m,m'} \, \hat{V}_{n-n'}. \tag{G.29}$$

Taking the squared magnitude of Eq. G.29, we obtain:

$$|C_{(m,n),(m',n')}|^2 = k^4_{m',n'} \, \delta_{m,m'} \, |\hat{V}_{n-n'}|^2. \tag{G.30}$$

To compute the Frobenius norm, we sum Eq. G.30 over all indices $(m, n, m', n')$:

$$\|C\|^2_F = \sum_{m,n} \sum_{m',n'} |C_{(m,n),(m',n')}|^2 = \sum_m \sum_{n,n'} k^4_{m,n'} \, |\hat{V}_{n-n'}|^2. \tag{G.31}$$

We now apply Parseval's identity to simplify the Fourier coefficient sum in Eq. G.31. Defining $q = n - n'$ and noting that $\hat{V}_q$ are the Fourier coefficients of $f(y) = \delta c^2(y)$, we have:

$$\sum_q |\hat{V}_q|^2 = \frac{1}{L_y} \int_0^{L_y} |\delta c^2(y)|^2 \, dy. \tag{G.32}$$

Substituting Eq. G.32 into Eq. G.31 gives us:

$$\|C\|^2_F \propto \left( \sum_{m,n'} k^4_{m,n'} \right) \cdot \int_0^{L_y} |\delta c^2(y)|^2 \, dy. \tag{G.33}$$

To maximize Eq. G.33 under the constraints $c^2_{\min} \leq c^2 \leq c^2_{\max}$ with fixed mean, we need to maximize the integral $\int_0^{L_y} |\delta c^2(y)|^2 \, dy$. This quantity is maximized when $\delta c^2(y)$ attains only the extreme values $\{c^2_{\min} - c^2_0, \; c^2_{\max} - c^2_0\}$. This follows from the convexity of $x \mapsto x^2$: any intermediate values reduce the integral compared to two-level allocations with the same mean.

Therefore, binary-valued striping maximizes $\|\delta c^2\|_{L^2}$ and thus the Frobenius norm $\|C\|^2_F$, yielding maximal total coupling strength across modes. $\qquad\square$

**Lemma 6** (Target Mode-Pair Coupling Maximization). *For a target mode pair $(m, n) \leftrightarrow (m, n + q_0)$, the coupling coefficient is*

$$|C_{(m,n),(m,n+q_0)}| = k^2_{m,n+q_0} \, |\hat{V}_{q_0}|, \tag{G.34}$$

*where $\hat{V}_{q_0}$ is the $q_0$-th Fourier coefficient of $\delta c^2(y)$. Under amplitude bounds and a fixed mean, the maximizer of $|\hat{V}_{q_0}|$ is a binary two-level function in y aligned with the $q_0$-Fourier kernel, i.e. a binary striped pattern whose stripe period matches $q_0$.*

*Proof.* We begin by applying the selection rule from Lemma 4, which shows that for striped perturbations:

$$C_{(m,n),(m',n')} = k^2_{m',n'} \, \delta_{m,m'} \, \hat{V}_{n-n'}. \tag{G.35}$$

Fixing $(m', n') = (m, n + q_0)$ in Eq. G.35, we obtain:

$$|C_{(m,n),(m,n+q_0)}| = k^2_{m,n+q_0} \, |\hat{V}_{q_0}|. \tag{G.36}$$

The Fourier coefficient $\hat{V}_{q_0}$ appearing in Eq. G.36 is defined as:

$$\hat{V}_{q_0} = \frac{1}{L_y} \int_0^{L_y} \delta c^2(y) \, e^{i \frac{2\pi q_0 y}{L_y}} \, dy. \tag{G.37}$$

From Eq. G.37, we see that $|\hat{V}_{q_0}|$ represents the magnitude of the inner product between $\delta c^2(y)$ and the Fourier kernel $e^{i 2\pi q_0 y / L_y}$.

To maximize the coupling strength in Eq. G.36, we seek to maximize $|\hat{V}_{q_0}|$ subject to the constraints:

$$c^2_{\min} - c^2_0 \; \leq \; \delta c^2(y) \; \leq \; c^2_{\max} - c^2_0, \qquad \frac{1}{L_y} \int_0^{L_y} \delta c^2(y) \, dy = \text{const.} \tag{G.38}$$

This optimization problem involves maximizing a linear functional of $\delta c^2$ over a convex feasible set defined by Eq. G.38. By standard convexity arguments, the maximum is attained at an extremal point of the admissible set, which means $\delta c^2(y)$ takes only the extreme values $\{c_{\min}^2 - c_0^2, c_{\max}^2 - c_0^2\}$ almost everywhere. The optimal arrangement is given by thresholding along the sign of $\Re(e^{i2\pi q_0 y/L_y})$, effectively aligning with the $q_0$-Fourier kernel.

Therefore, the optimal $\delta c^2$ is a binary stripe pattern in $y$ with period $L_y/q_0$. This configuration yields maximal selective coupling between modes $(m, n)$ and $(m, n + q_0)$. $\qquad\square$

## H WAVE SPEED DISTRIBUTION CALCULATION

### H.1 BIOOSS WAVE SPEED FORMULA IMPLEMENTATION

The spatially-varying wave speed $c(x, y)$ in our BioWM implementation is calculated using the BioOSS eigenfrequency formula from Theorem 1. For each grid location corresponding to PANN dimension $i$, we compute:

$$c_i = \frac{\tan(\pi f_i \Delta t)\sqrt{(1 + \Delta t k_p)(1 + \Delta t k_v)}}{\Delta t \sqrt{2}}, \tag{H.1}$$

where $f_i$ is the target frequency for PANN dimension $i$, $\Delta t = 0.01$ (time step), $k_p = k_v = 10.0$ (damping coefficients), and the factor $\sqrt{2}$ accounts for 2D isotropic spatial discretization.

### H.2 FREQUENCY-TO-GRID MAPPING

Target frequencies are linearly distributed across PANN dimensions:

$$f_i = f_{\min} + (f_{\max} - f_{\min}) \times \frac{i}{526}, \quad i \in \{0, 1, \ldots, 526\} \tag{H.2}$$

with $f_{\min} = 50\,\text{Hz}$ and $f_{\max} = 1200\,\text{Hz}$ in our experiments.

Grid positions are allocated by dividing the $64 \times 64$ lattice among the 527 PANN dimensions:

$$\text{positions per PANN} = \left\lfloor \frac{4096}{527} \right\rfloor \approx 7\text{–}8 \text{ positions per dimension} \tag{H.3}$$

### H.3 NUMERICAL STABILITY CONSTRAINTS

Two constraints ensure numerical stability:

**Upper bound** (CFL-like condition):

$$c_{\max} = \frac{1}{\Delta t \sqrt{2}}\sqrt{(1 + \Delta t k_p)(1 + \Delta t k_v)} \approx 77.8 \tag{H.4}$$

$$c_i = \min(c_i, 0.9 \times c_{\max}) \approx 70.0 \tag{H.5}$$

**Lower bound** (avoiding stagnation):

$$c_i = \max(c_i, 0.1) \tag{H.6}$$

The resulting wave speed field exhibits the stripe-like binary distribution predicted by our theoretical analysis.

## I ADDITIONAL QUALITATIVE COMPARISONS

To complement the illustrative cases in Section 3.3, we provide six additional Mel-spectrogram comparisons between BioWM and the cosine similarity baseline. These examples highlight diverse acoustic scenarios and further demonstrate the relative robustness of BioWM.

- **Car horn onset (Example R0003).** As shown in Fig. I.1, a car horn appears at 21 s and persists for several seconds. Both methods successfully detect the onset, but BioWM localizes the event with fewer spurious triggers.

- **Intermittent piano playing (Example R0007).** As shown in Fig. I.2, piano notes occur intermittently, producing pauses between onsets. BioWM registers four discrete activations, whereas cosine similarity generates numerous false alarms due to continuous drift sensitivity.

- **Conversational speech (Example R0028).** As shown in Fig. I.3, after 30 s, a dialogue with pauses and speaker changes begins. BioWM detects a single event at 38 s, while cosine similarity produces many detections starting at 31 s, reflecting its oversensitivity to intra-speech variability.

- **Repeated car horn sounds (Example R0030).** As shown in Fig. I.4, horns occur around 24 s, 32 s, and 50 s. BioWM detects only the late instance (around 51 s), whereas cosine similarity captures only the earlier two, underscoring differences in temporal selectivity.

- **Railway station announcement (Example R0031).** As shown in Fig. I.5, the segment contains continuous speech without other salient events. BioWM triggers only once at the beginning during threshold adaptation, while cosine similarity repeatedly fires between 29–48 s due to speech sensitivity.

- **Church bell with crowd talking (Example R0131).** As shown in Fig. I.6, continuous bell ringing dominates the spectrogram. Cosine similarity detects background crowd talking, but BioWM's adaptive threshold is raised by the persistent bell, suppressing these detections.

Taken together, these additional comparisons reinforce the findings from the main paper: BioWM avoids excessive drift detections in the presence of transient pauses, repeated motifs, or continuous sources, while maintaining sensitivity to salient novel events. Cosine similarity, by contrast, tends to over-trigger in response to spectral fluctuations within ongoing sound streams.

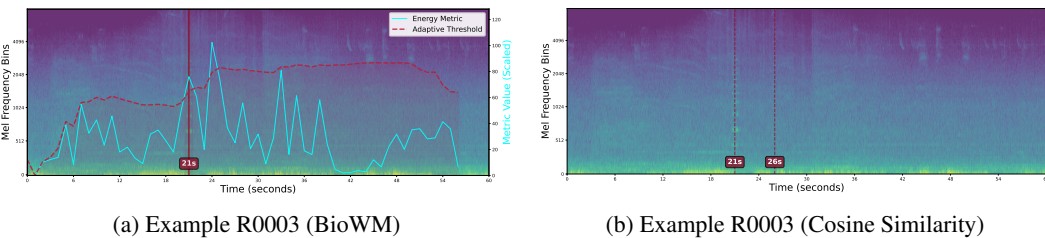

(a) Example R0003 (BioWM)     (b) Example R0003 (Cosine Similarity)

Figure I.1: Example R0003 (Tour de l'horloge, Montreal — park): Both approaches successfully detect the prominent novel event, the onset of a car horn around 21 s, which persists for several seconds.

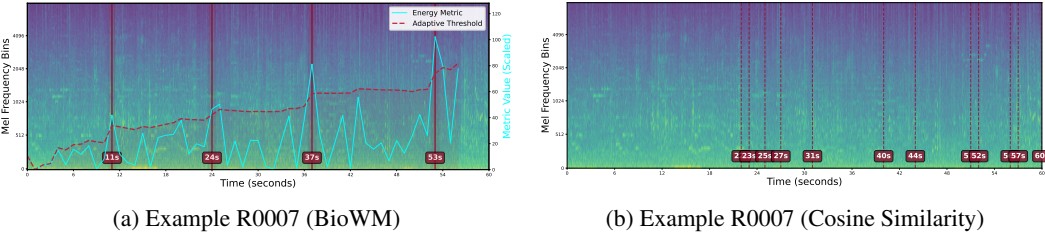

(a) Example R0007 (BioWM)     (b) Example R0007 (Cosine Similarity)

Figure I.2: Example R0007 (Chalet du Mont-Royal, Montreal — street traffic): The most salient event is intermittent piano playing, which triggers BioWM four times but produces many continuous drift alarms when using cosine similarity.

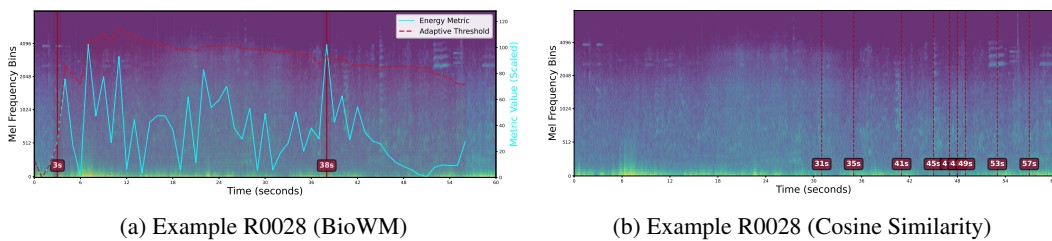

(a) Example R0028 (BioWM)       (b) Example R0028 (Cosine Similarity)

Figure I.3: Example R0028 (Heping Road, Tianjin — street traffic): The most salient event is conversation after 30 s, with pauses and speaker changes. BioWM is triggered only once at 38 s, whereas cosine similarity produces many continuous drift alarms starting from 31 s.

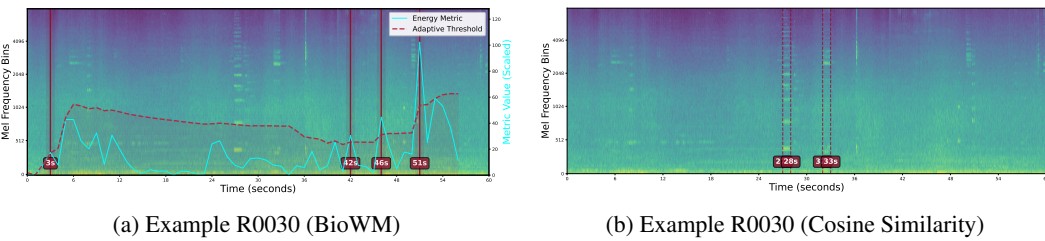

(a) Example R0030 (BioWM)       (b) Example R0030 (Cosine Similarity)

Figure I.4: Example R0030 (Century Clock, Tianjin — street traffic): The most salient event is repeated car horn sounds at approximately 24 s, 32 s, and 50 s. BioWM detects only the instance around 51 s, while cosine similarity detects only those around 24 s and 32 s.

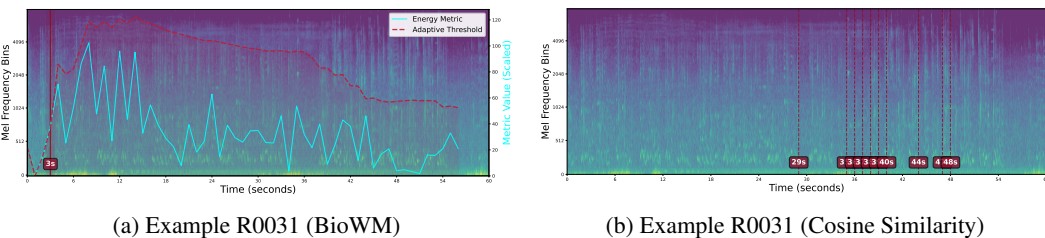

(a) Example R0031 (BioWM)       (b) Example R0031 (Cosine Similarity)

Figure I.5: Example R0031 (Tianjin Railway Station, Tianjin — public square): The entire audio segment is a railway station announcement without other salient events. BioWM is triggered only once at the beginning during threshold adaptation, whereas cosine similarity is highly sensitive to speech, producing many detections between 29 s and 48 s.

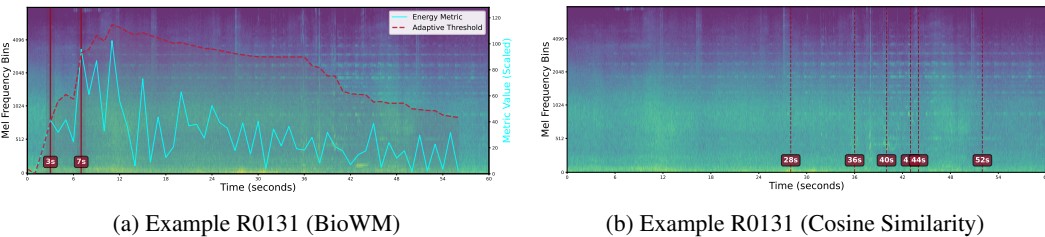

(a) Example R0131 (BioWM)       (b) Example R0131 (Cosine Similarity)

Figure I.6: Example R0131 (Town Hall Square, Vilnius — square): The entire audio segment contains continuous church bell ringing, clearly visible on the Mel spectrogram. Cosine similarity captures several crowd-talking events in the square, but the persistent bell raises the BioWM threshold, preventing detection of the crowd talking.

## J   ADDITIONAL FFT ANALYSIS

In the main paper (Section 3.4), we presented FFT analyses for Examples R0016 and R0056 to illustrate how BioWM reallocates oscillatory activity around drift onsets. To provide further evidence, Fig. J.1 shows additional FFT spectra for five representative cases (R0002, R0010, R0016, R0037, R0056). These examples span a variety of acoustic scenes, including novel sound events (R0002), subcategory-level drift (R0010), transient pauses (R0016, R0037), and salient novel sources (R0056).

Across all examples, the dominant oscillatory activity of $p$-field neurons lies within the $\theta$ (4–8 Hz), $\alpha$ (8–12 Hz), $\beta$ (13–30 Hz), and low-$\gamma$ (30–50 Hz) bands, consistent with canonical neural regimes. Rather than being uniformly distributed, activity is organized in clustered regions that evolve after drift onsets. The figures display the activity of all $p$ neurons over entire segments; thus, direct saliency patterns are not immediately visible. What becomes evident instead is the periodic coupling structure along the Y-axis of the lattice, in line with the striped optimality predicted by Theorem 5. This suggests that BioWM's drift sensitivity arises from frequency-specific clustering combined with spatial coupling, rather than from broad or diffuse spectral fluctuations. Animated visualizations of these dynamics are included as supplementary material.

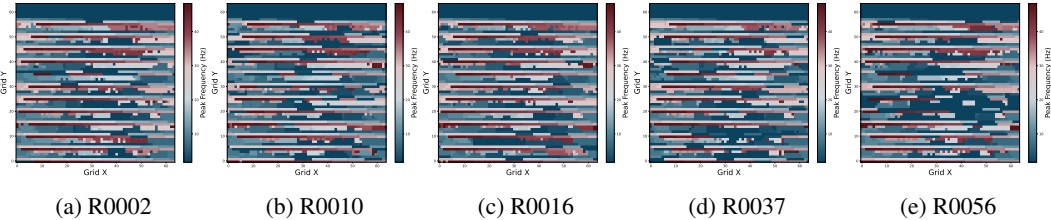

(a) R0002          (b) R0010          (c) R0016          (d) R0037          (e) R0056

Figure J.1: Frequency distribution across the $p$-field neurons obtained via FFT for different Examples (R0002, R0010, R0016, R0037, R0056). The results reveal band-specific activity in the theta (4–8 Hz), alpha (8–12 Hz), beta (13–30 Hz), and gamma (30–50 Hz) ranges.

## K   APPENDIX FOR DRIFT DETECTION RATE (DDR)

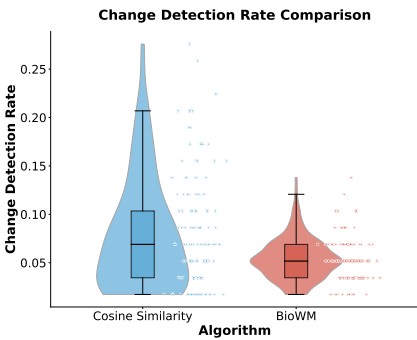

Figure K.1: Drift detection rate comparison between BioWM and the cosine similarity baseline. BioWM achieves a markedly lower detection rate, reflecting its superior ability to capture genuine auditory drift events while minimizing spurious detections.

Evaluating online auditory drift detection poses unique challenges because no benchmark dataset with ground-truth drift annotations is currently available. In this context, the Drift Detection Rate (DDR) serves as a practical and informative proxy metric. DDR is defined as the normalized frequency of detected drifts within an audio segment. It complements qualitative case studies by providing a quantitative summary of detector behavior across diverse soundscapes. Importantly, DDR captures the trade-off between sensitivity and robustness: a detector with excessively high DDR risks over-triggering on natural variability, while an overly low DDR indicates missed salient changes. By reporting DDR, we can systematically compare different detection methods on a common scale and

assess whether a model consistently avoids false positives while remaining responsive to meaningful auditory events. Thus, even in the absence of ground truth labels, DDR offers a valuable measure of calibration and stability for unsupervised drift detection systems.

Note that the first 15 s of each audio segment are always forwarded to the higher cognition module regardless of drift detection. Therefore, when computing the DDR, we only count detections occurring after the initial 15 s period. The final DDR value is obtained by normalizing the number of post-15 s detections by the segment duration and then adding one to account for the mandatory transmission of the initial 15 s window:

$$\text{DDR} = \frac{N_{\text{drift}}^{(>15\,s)}}{T_{\text{audio}}} + 1, \tag{K.1}$$

where $N_{\text{drift}}^{(>15\,s)}$ denotes the number of detected drifts after the first 15 s, and $T_{\text{audio}}$ is the total duration of the audio segment.

**Qwen outputs.** To further illustrate the downstream impact, we provide representative drift descriptions generated by Qwen when conditioned on BioWM detections versus cosine similarity triggers. Qwen produces coherent and contextually accurate summaries when driven by BioWM signals, whereas cosine similarity often results in fragmented or spurious descriptions. Full transcripts are attached in the supplementary materials.

## L  QUANTITATIVE EVIDENCE FOR THE NECESSITY OF NAACA IN ALM

To assess the temporal memory abilities of Audio Qwen, a representative ALM, we evaluate its sound-event detection accuracy on a synthetic one-minute recording composed of twelve distinct clips, each featuring a different event. This controlled setup enables a quantitative assessment of temporal bias. Next, we examine a real-world dataset and select a representative example to qualitatively demonstrate how late-occurring salient events can be suppressed due to information compression and attention imbalance. Finally, we evaluate the performance gains achieved by our proposed cognitively inspired framework, which selectively enhances salient segments through event-driven attention gating, while also demonstrating improved computational efficiency compared to using Qwen as the monitoring mechanism.

### L.1  TEMPORAL MEMORY CAPABILITY ANALYSIS

We evaluated Audio Qwen's temporal attention limitations using the ESC-50 dataset by concatenating 12 distinct 5-second clips (each containing a single audio event like dog barking or rain) into 60-second sequences. Each temporal position (0-5s, 5-10s, ..., 55-60s) contained a ground-truth event, and the model was queried across 500 independent trials to identify and order sound events.

The results showed a marginally significant temporal decay in recognition accuracy (Fig. L.1), with performance decreased from 0.35 (10-20s peak) to 0.06 (50s), representing a substantial relative decrease. Linear regression analysis revealed a negative correlation between temporal position and accuracy ($R^2 = 0.293$, $p = 0.069$, slope $= -0.0141$). Although the $p$-value exceeds the conventional significance threshold, the consistent trend across 500 trials indicates temporal limitations in the model's attention mechanism. Performance drops sharply around 35 seconds, marking the boundary for effective attention. This transition supports an optimal attention window of $\sim$35 seconds, beyond which the model struggles to recall audio events. These findings suggest that for audio sequences longer than 35 seconds, the model may miss critical contextual cues. This limitation is particularly important for tasks such as meeting transcription, long-form content analysis, or temporal understanding about audio events.

### L.2  COMPUTATIONAL ANALYSIS

We further evaluated the computational efficiency of our proposed framework in real-time monitoring scenarios. Consider a real-time monitoring task with a 1-minute duration. When we used Audio Qwen as the pattern-changing detector in the ALM framework, the response for each 4-second sliding window took approximately 0.5 seconds. However, by incorporating echoic memory

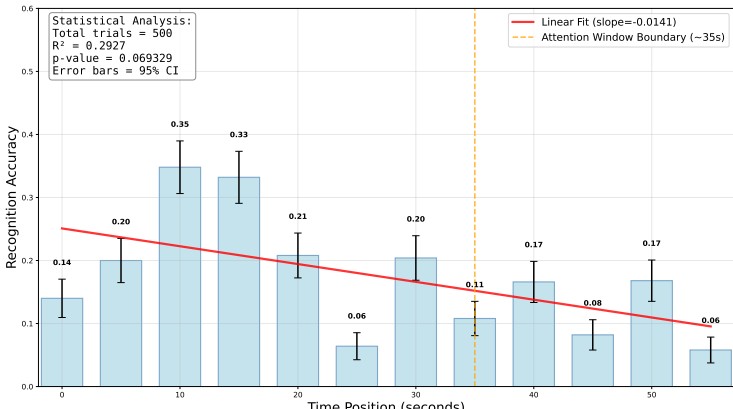

Figure L.1: **Temporal decay in audio event recognition.** Recognition accuracy of Audio Qwen across 12 temporal positions in 60-second composite audio sequences from ESC-50 dataset Piczak (2015). Each bar represents the mean accuracy with 95% confidence intervals from 500 total trials. The red line shows the linear regression fit ($R^2 = 0.293$, $p = 0.069$), indicating a marginally significant temporal decay pattern. The orange dashed line indicates the critical attention window boundary at approximately 35 seconds, beyond which performance degrades substantially. Peak performance occurs in the 10-20 second range (0.33-0.35 accuracy), while performance drops to 0.06-0.17 in later segments (40-50s), demonstrating the model's limited temporal attention span for extended audio sequences.

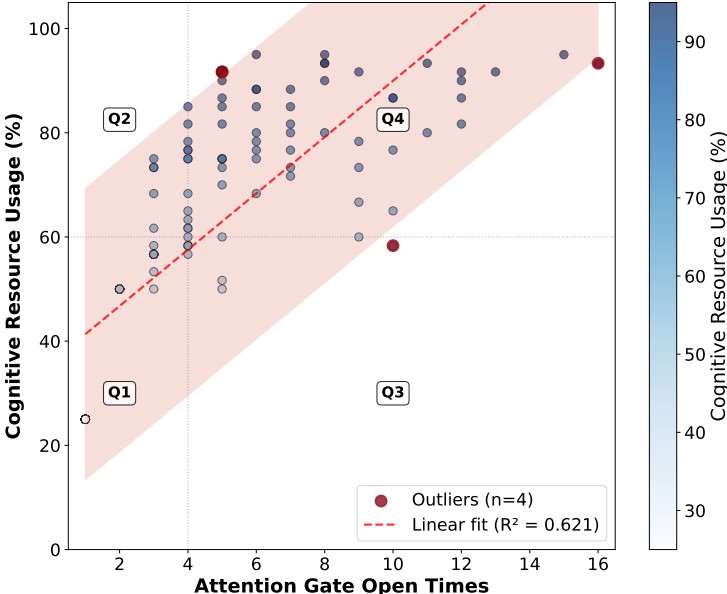

Figure L.2: **Attention gate activations vs. cognitive resource usage.** Each point represents a USoW dataset sample. X-axis: attention gate open times; Y-axis: cognitive resource usage (%) - ratio of Audio Qwen processed sequence length. Light red area shows 95% confidence interval. Quadrants (Q1-Q4) divide samples by median values. Red circles indicate outliers.

as a monitor, the cosine similarity calculation is reduced to only around 0.02 seconds, resulting in a significant 96% reduction in computation time.

The 15-second attention maintenance window is sent to Audio Qwen for response generation, serving as a trade-off solution between the powerful performance of the ALM framework and computa-

tional efficiency. We calculated the attention gate open times for each sample in the USoW dataset to determine the total processing time for each 1-minute interval and visualize these results in Fig. L.2.

Our analysis in Fig. L.2 revealed a significant positive correlation between attention gate open times and cognitive resource usage ($R^2 = 0.621$, linear fit shown in red dashed line). The quadrant analysis demonstrated four distinct operational regimes based on median splits of both variables (median attention gates = 4). The Q1 quadrant represents stable audio segments with minimal change, while Q4 indicates intense changes distributed throughout the segment, representing common operational cases. The Q2 quadrant (low gate opening, high resource usage) captures segments where changes are evenly distributed, requiring high computational resources despite fewer attention triggers. Most notably, the Q3 quadrant represents an optimal operational state where changes are densely concentrated within specific attention windows, allowing the system to maintain high vigilance while achieving computational efficiency. The median attention gate activation of 4 times per segment demonstrates substantial computational savings, with Audio Qwen being triggered only 4 times rather than the full 60 times per 1-minute segment, achieving a 93.33% reduction in computational time compared to continuous monitoring.

Four outliers (n=4, highlighted in red) deviate substantially from the general linear trend, suggesting exceptional cases that may reflect unique audio patterns or system states requiring further examination. For two outliers, R0074 and R0096, the salient events are sparse. In contrast, R0016 and R0090 exhibit more complex temporal structures. R0016 is analysed in detail in Fig. 3b. Likewise, in R0090 (Fig. L.3), the cosine–similarity detector is frequently triggered during the final 20 s of the recording. Because the spectrogram varies gradually, the cosine similarity remains consistent during the first 40 s and stays below the threshold toward the end. Our method, however, is designed to be more sensitive to semantic events rather than spectral similarity alone, which explains its robustness to gradual spectrogram variation.

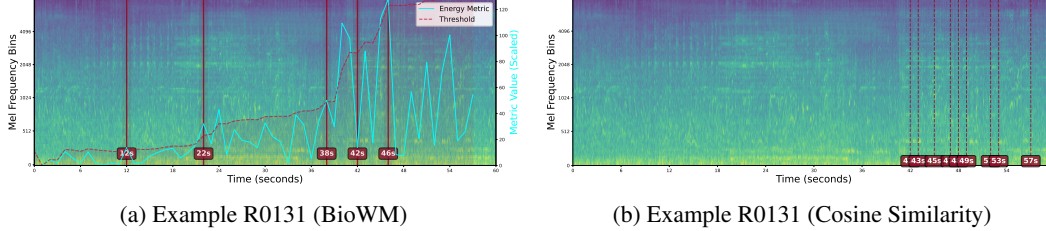

(a) Example R0131 (BioWM)        (b) Example R0131 (Cosine Similarity)

Figure L.3: Example R0090 (Millennium, Chicago — square): The audio segment contains conversational speech during the first 18 s, followed by music that begins at approximately 18 s and ends around 29 s. In the final 20 s, background talking and singing cause rapid, second-by-second fluctuations in the spectrogram, which in turn lead to strong variability in the cosine similarity values. These fluctuations are effectively handled by the BioWM detector.

## M    FULL EVENT DETECTION COMPARISON

**Experimental Environment.** All experiments were implemented using PyTorch 2.6.0+cu124 with CUDA 12.4 support, within a Python 3.9.13 environment. The codebase was executed on a Linux system equipped with a single NVIDIA RTX 4090 GPU (24 GB VRAM), an Intel Core i9 CPU, and 128 GB of RAM. Audio preprocessing was performed using `torchaudio` and `librosa`, and inference was carried out using frozen pretrained models including Audio Qwen and PANN (CNN14). Random seeds were fixed where applicable to ensure reproducibility across runs.

**Experimental Setup.** To evaluate the effectiveness of our proposed cognitive attention framework in capturing salient auditory events over extended durations, we compared it against two baselines: the PANN and the pretrained Audio Qwen model. Each model was applied to the same set of

133 one-minute audio samples selected from the USoW dataset, covering diverse real-world urban soundscapes. For PANN, we used the pretrained CNN14 model trained on AudioSet to perform sound event detection and retained only the events with posterior probability above 0.4. If no event in a sample exceeded this threshold, the top-scoring event was selected. The Audio Qwen baseline processed the full 60-second segment in a single forward pass without segmentation or temporal attention adjustment. In contrast, our framework dynamically routed only semantically salient 15-second windows—identified through pattern-change detection—into the same frozen Audio Qwen model for interpretation. This ensured a fair comparison across all models while minimizing inference cost and preserving model architecture integrity.

**Results Summary.** The comparative results, shown in Table M.1, demonstrate the clear advantages of our cognitively inspired framework in identifying temporally distributed and contextually significant sound events. While PANN primarily detects isolated high-confidence events and often overlooks complex auditory scenes, Audio Qwen exhibits a notable tendency to miss late-occurring or weakly segmented sounds due to attention decay over long sequences. Our framework consistently captured more events per sample and provided richer acoustic descriptions by selectively focusing on semantically important windows. For example, events such as bagpipes, crowd reactions, or musical instruments—frequently missed by both baselines—were successfully identified in the cognitive setting. This not only confirms the theoretical analysis of temporal bias in ALMs but also highlights the practical benefits of our system for long-form audio understanding, with improved granularity and coverage of diverse environmental cues.

Table M.1: Comparison of events detected by PANN, Qwen, and our Hierarchical Selective Attention (HSA) framework.

| File Index | PANN Detected Events | Qwen Detected Events | HSA Detected Events |
|---|---|---|---|
| R0001 | rain: 0.359 | construction; rain | metal; rain |
| R0002 | vehicle: 0.419 | birds; car; door; traffic | birds; car; speech; traffic; wind |
| R0003 | vehicle: 0.397 | birds; rain; traffic; train; train_whistle | birds; car; machinery; rain; traffic; train; train_whistle |
| R0004 | speech: 0.526 | birds; speech; traffic | birds; horn; leaves; rain; speech; traffic; train; train_whistle; wind |
| R0005 | animal: 0.228 | birds; children; construction; speech | airplane; birds; dog; traffic; wind |
| R0006 | vehicle: 0.571 | machinery | birds; car; horn; speech; traffic; wind |
| R0007 | speech: 0.841; music: 0.402 | footsteps; music; speech | basketball; bell; birds; car; children; crowd; engine; footsteps; laughter; music; speech; telephone; whistle |
| R0008 | vehicle: 0.179 | birds; car; speech; traffic | footsteps; laughter; leaves; music; outdoor; speech; traffic |
| R0009 | vehicle: 0.424 | birds; traffic; ventilation | birds; traffic |
| R0010 | speech: 0.666; music: 0.527 | car; laughter; music; speech; street; traffic | birds; car; horn; machinery; music; park; speech; street; traffic; urban |
| R0011 | speech: 0.591 | bell; birds; speech; telephone | bell; birds; footsteps; horn; speech; telephone; traffic |
| R0012 | vehicle: 0.524; speech: 0.508 | helicopter | airplane; birds; bus; construction; crowd; footsteps; laughter; restaurant; speech; street; traffic; ventilation |
| R0013 | vehicle: 0.546 | airplane; birds | airplane; birds; traffic; wind |
| R0014 | vehicle: 0.517 | birds; speech; wind | birds; wind |
| R0015 | speech: 0.714 | birds; car; speech; traffic | car; footsteps; horn; singing; speech; traffic; trumpet |
| R0016 | speech: 0.828 | machinery | birds; cheering; clapping; crowd; footsteps; laughter; speech; telephone |

| File Index | PANN Detected Events | Qwen Events | Cognitive Events |
|---|---|---|---|
| R0017 | speech: 0.687 | machinery | birds; car; engine; laughter; speech; traffic; water |
| R0018 | vehicle: 0.574; speech: 0.454 | construction; speech; traffic | car; children; horn; speech; traffic; whistle |
| R0019 | speech: 0.538 | wind | construction; laughter; rain; speech; train; train_whistle; whistle |
| R0020 | music: 0.346 | crying; speech; traffic | alarm; birds; car; rain; traffic; train; train_whistle; whistle |
| R0021 | speech: 0.589 | birds; traffic | birds; car; traffic; wind |
| R0022 | speech: 0.554 | children; leaves; rain | car; construction; horn; rain; thunder; wind |
| R0023 | speech: 0.694 | birds; car; engine; speech | birds; car; children; laughter; rain; speech; traffic; train; train_whistle; whistle |
| R0024 | animal: 0.314 | birds; insects; speech | birds; insects; singing |
| R0025 | thunder: 0.137 | rain; train | airplane; birds; traffic |
| R0026 | speech: 0.609; music: 0.564 | car; music; speech; traffic | car; music; speech; traffic |
| R0027 | speech: 0.382 | children; leaves; wind | birds; children; leaves; speech; water; wind |
| R0028 | speech: 0.741 | children; speech; wind | ambulance; birds; car; children; horn; laughter; speech; traffic; trumpet; whistle; wind |
| R0029 | vehicle: 0.607; train: 0.562; vehicle horn, car horn, honking: 0.487; rail transport: 0.419; train horn: 0.412 | rain; train; train_whistle | bell; birds; car; rain; telephone; traffic; train; train_whistle |
| R0030 | vehicle: 0.594; toot: 0.441 | machinery; rain; train | ambulance; car; horn; speech; traffic |
| R0031 | speech: 0.769 | machinery | rain; speech; telephone; train |
| R0032 | speech: 0.609 | children; speech; traffic | birds; car; children; engine; laughter; motorcycle; speech; traffic |
| R0033 | vehicle: 0.667; toot: 0.445; speech: 0.415 | birds; construction; traffic | birds; car; horn; speech; traffic |
| R0034 | speech: 0.707 | birds; children; speech; traffic | birds; camera; car; children; engine; horn; laughter; speech; traffic |
| R0035 | speech: 0.627 | children; speech; traffic | car; children; laughter; speech; traffic |
| R0036 | vehicle: 0.491 | birds; construction; traffic | birds; car; construction; horn; traffic |
| R0037 | speech: 0.631 | birds; speech; traffic | birds; car; children; crying; screaming; shouting; speech; traffic |
| R0038 | vehicle: 0.58 | birds; construction; traffic | birds; car; traffic |
| R0039 | speech: 0.352 | birds; car; children; engine | birds |
| R0040 | environmental noise: 0.283 | birds; car; rain; traffic; train | birds |
| R0041 | speech: 0.713 | footsteps; rain; speech; train | car; speech; traffic |
| R0042 | speech: 0.661 | construction; speech; traffic | birds; bus; car; cat; construction; crowd; footsteps; horn; insects; laughter; rain; speech; street; traffic; train; train_whistle; urban; whistle |
| R0043 | vehicle: 0.36 | birds; construction; speech; traffic | birds; traffic |
| R0044 | vehicle: 0.314 | birds; car; horn; speech; traffic; wind | traffic |
| R0045 | speech: 0.739 | birds; car; speech; traffic | birds; car; children; crowd; engine; laughter; speech; traffic |
| R0046 | speech: 0.662 | birds; construction; speech | birds; car; construction; crowd; engine; laughter; machinery; metal; speech; traffic; truck |

| File Index | PANN Detected Events | Qwen Events | Cognitive Events |
|---|---|---|---|
| R0047 | vehicle: 0.696 | speech; traffic | car; horn; rain; speech; traffic; train; train_whistle |
| R0048 | speech: 0.421 | birds; car; speech; traffic | birds; footsteps; machinery |
| R0049 | speech: 0.646 | birds; car; engine; speech | birds; children; footsteps; speech; traffic |
| R0050 | speech: 0.335 | music; speech; traffic | car; horn; speech; traffic; trumpet; wind |
| R0051 | animal: 0.488 | birds; crying; speech | birds; crying; footsteps; outdoor |
| R0052 | speech: 0.458 | airplane; birds; traffic | airplane; birds; car; traffic |
| R0053 | vehicle: 0.559; field recording: 0.412 | traffic | traffic |
| R0054 | speech: 0.635 | car; footsteps; laughter; speech; traffic; wind | bell; bicycle; bicycle_bell; birds; car; engine; laughter; motorcycle; speech; traffic |
| R0055 | speech: 0.833 | car; speech; traffic; water | camera; car; footsteps; speech |
| R0056 | speech: 0.511; music: 0.415 | speech; water | bagpipe; birds; leaves; music; rain; speech; wind |
| R0057 | speech: 0.553 | birds; speech; water | birds; children; speech; water |
| R0058 | music: 0.652; speech: 0.539 | singing; street | birds; bus; cat; crowd; door; drums; guitar; laughter; music; outdoor; piano; singing; speech; water; wind |
| R0059 | speech: 0.646 | birds; rain; speech; train | basketball; birds; car; engine; laughter; music; ocean; speech; traffic |
| R0060 | speech: 0.749; music: 0.647 | music; speech; traffic | car; cheering; crowd; engine; footsteps; laughter; music; singing; speech; street; telephone; traffic |
| R0061 | speech: 0.787 | nan | birds; children; footsteps; laughter; speech |
| R0062 | speech: 0.763 | shouting; speech | basketball; birds; speech |
| R0063 | vehicle: 0.686 | speech; traffic | bell; car; horn; speech; telephone; traffic |
| R0064 | speech: 0.704; vehicle: 0.584 | rain; speech; traffic; train | birds; car; construction; horn; machinery; rain; speech; traffic; train; train_whistle; whistle |
| R0065 | vehicle: 0.574 | traffic | birds; construction; machinery; traffic |
| R0066 | speech: 0.658; vehicle: 0.521 | speech; traffic | birds; car; footsteps; speech; traffic; wind |
| R0067 | speech: 0.767 | machinery | birds; footsteps; laughter; speech; traffic; wind |
| R0068 | speech: 0.66 | airplane; birds; car; traffic | birds; car; footsteps; speech; traffic |
| R0069 | speech: 0.634; vehicle: 0.441 | birds; car; horn; insects; laughter; speech; traffic | traffic; waves |
| R0070 | speech: 0.504; vehicle: 0.425 | car; speech; traffic; wind | bell; car; telephone; traffic |
| R0071 | music: 0.798; speech: 0.75 | footsteps; music; street; traffic | birds; cheering; clapping; crowd; laughter; music; speech; street; telephone; traffic; trumpet |
| R0072 | speech: 0.716 | door; footsteps; metal; speech | birds; car; door; engine; footsteps; horn; rain; speech; traffic; train; train_whistle; whistle |
| R0073 | speech: 0.617; emergency vehicle: 0.412 | ambulance; speech; traffic | ambulance; car; crowd; engine; footsteps; laughter; music; speech; street; telephone; traffic |
| R0074 | speech: 0.728; vehicle: 0.64 | birds; laughter; ocean; restaurant; speech; traffic | birds; car; children; crying; engine; horn; laughter; music; speech; traffic |
| R0075 | speech: 0.724 | birds; speech; traffic | birds; car; rain; speech; traffic; train |

| File Index | PANN Detected Events | Qwen Events | Cognitive Events |
|---|---|---|---|
| R0076 | speech: 0.644 | children; speech; traffic | birds; children; footsteps; speech; traffic; water |
| R0077 | emergency vehicle: 0.598; police car (siren): 0.597; speech: 0.477; siren: 0.466 | machinery | ambulance; cheering; children; footsteps; laughter; speech; telephone; traffic |
| R0078 | rain: 0.351 | birds; car; horn; ocean; rain; speech; traffic; urban | ocean; rain; traffic |
| R0079 | speech: 0.462 | birds; car; engine; speech | airplane; birds; engine; machinery |
| R0080 | speech: 0.644 | birds; car; speech; traffic | bell; bicycle; bicycle_bell; birds; camera; car; footsteps; insects; speech; traffic; whistle |
| R0081 | music: 0.761; speech: 0.545 | machinery | children; music; ocean; piano; singing; speech; waves |
| R0082 | speech: 0.605 | car; speech; traffic | car; children; engine; speech; traffic |
| R0083 | vehicle: 0.339 | birds; speech; traffic | ambulance; bell; birds; car; crying; footsteps; horn; laughter; outdoor; rain; speech; telephone; traffic; train; train_whistle; truck; whistle |
| R0084 | vehicle: 0.728; speech: 0.682; air brake: 0.442 | speech; traffic | bell; bicycle; birds; car; engine; horn; rain; speech; telephone; traffic; train; train_whistle; truck; wind |
| R0085 | speech: 0.727; vehicle: 0.449 | machinery | birds; car; children; laughter; machinery; metal; speech; traffic; truck |
| R0086 | speech: 0.624 | ambulance; birds; bus; cat; children; crowd; horn; machinery; music; park; rain; restaurant; speech; traffic; train; urban | ambulance; birds; boat; bus; car; children; crowd; engine; horn; laughter; speech; street; telephone; traffic; waves; wind |
| R0087 | speech: 0.842 | machinery; rain; train | ambulance; children; laughter; music; speech; trumpet; violin |
| R0088 | vehicle: 0.634; speech: 0.621 | birds; speech; traffic | car; crowd; engine; laughter; speech; traffic |
| R0089 | speech: 0.778 | cat; footsteps; laughter; leaves; music; outdoor; speech; traffic | car; crowd; laughter; rain; speech; traffic; train; train_whistle; whistle |
| R0090 | speech: 0.751; music: 0.687 | footsteps; music; speech | cheering; clapping; crowd; footsteps; horn; laughter; music; singing; speech; street; telephone; trumpet |
| R0091 | music: 0.476; speech: 0.402 | crowd; telephone | camera; cat; cheering; clapping; crowd; laughter; music; shouting; singing; speech; telephone; trumpet |
| R0092 | speech: 0.713; vehicle: 0.469 | footsteps; speech; traffic | car; footsteps; rain; speech; traffic; train; wind |
| R0093 | vehicle: 0.301 | machinery | ventilation |
| R0094 | speech: 0.478 | birds; children; speech; traffic | birds; speech; traffic |
| R0095 | speech: 0.617; vehicle: 0.519 | footsteps; speech; traffic | car; rain; speech; traffic; train |
| R0096 | speech: 0.423 | children; speech; traffic | children; crowd; laughter; rain; shouting; speech; traffic; water |
| R0097 | crow: 0.646; caw: 0.615; animal: 0.457 | birds; traffic | birds; traffic |
| R0098 | police car (siren): 0.49; emergency vehicle: 0.433 | cat; machinery | ambulance; car; traffic |
| R0099 | speech: 0.373 | bell; birds; car; speech; traffic | bell; speech; telephone; traffic |
| R0100 | vehicle: 0.457 | engine; footsteps; speech | airplane; birds; cat; footsteps; laughter; music; speech; traffic; wind |
| R0101 | speech: 0.716; music: 0.601 | basketball; guitar; speech; sports | basketball; car; laughter; music; speech; traffic |

| File Index | PANN Detected Events | Qwen Events | Cognitive Events |
|---|---|---|---|
| R0102 | speech: 0.623; vehicle: 0.574 | birds; car; speech; traffic | birds; car; traffic |
| R0103 | speech: 0.828 | machinery | cheering; children; crowd; footsteps; laughter; music; shouting; speech; telephone |
| R0104 | speech: 0.801 | birds; car; speech; traffic | birds; car; clapping; engine; footsteps; laughter; speech; traffic |
| R0105 | speech: 0.868 | birds; laughter; speech | birds; laughter; speech |
| R0106 | speech: 0.798 | footsteps; speech; traffic | birds; car; footsteps; speech; traffic |
| R0107 | speech: 0.779; clip-clop: 0.47; horse: 0.469; animal: 0.459; music: 0.447 | bell; birds; speech; telephone | bell; birds; footsteps; music; speech; telephone; traffic |
| R0108 | speech: 0.581 | bus; crowd; footsteps; speech; traffic | birds; car; crowd; footsteps; speech; traffic; water; waves; wind |
| R0109 | speech: 0.813 | birds; car; speech; traffic | birds; car; footsteps; laughter; shouting; speech; traffic |
| R0110 | vehicle: 0.503 | birds; speech; traffic | birds; traffic |
| R0111 | vehicle: 0.452 | birds; traffic | birds; car; traffic |
| R0112 | speech: 0.866 | birds; footsteps; speech | birds; children; footsteps; laughter; machinery; speech |
| R0113 | speech: 0.786; animal: 0.643; horse: 0.476 | bell; birds; footsteps; telephone | birds; dog; rain; train; train_whistle; whistle; wind |
| R0114 | speech: 0.779 | birds; footsteps; speech | birds; car; children; crying; laughter; machinery; speech; traffic; water; wind |
| R0115 | speech: 0.657 | birds; children; shouting; speech | birds; children; laughter; screaming; shouting; speech |
| R0116 | vehicle: 0.654; speech: 0.546 | speech; traffic | birds; children; engine; motorcycle; speech; telephone; traffic |
| R0117 | speech: 0.884 | footsteps; laughter; speech; traffic | crowd; footsteps; laughter; machinery; music; speech; street |
| R0118 | music: 0.569; brass instrument: 0.534; speech: 0.454 | rain; train | birds; cheering; clapping; crowd; leaves; music; singing; telephone |
| R0119 | vehicle: 0.264 | birds; car; speech; traffic | traffic |
| R0120 | speech: 0.488; vehicle: 0.438 | birds; car; speech; street; traffic | alarm; birds; car; speech; traffic |
| R0121 | church bell: 0.737 | bell; telephone; traffic | bell; speech; telephone; traffic |
| R0122 | speech: 0.702 | birds; car; footsteps; speech; traffic | birds; car; crowd; footsteps; laughter; speech; traffic |
| R0123 | vehicle: 0.791; air brake: 0.457 | bus; engine; truck | birds; bus; car; crowd; speech; street; traffic |
| R0124 | speech: 0.714 | birds; car; engine; speech | airplane; birds; car; engine; footsteps; speech; traffic |
| R0125 | vehicle: 0.599 | car; telephone | car; speech; traffic |
| R0126 | speech: 0.27 | birds; water | crowd; water |
| R0127 | speech: 0.665; music: 0.433 | music; speech; traffic | bicycle; crowd; laughter; metal; music; restaurant; speech; street; traffic |
| R0128 | speech: 0.739 | construction; door; speech | birds; car; door; engine; footsteps; laughter; outdoor; speech; traffic |
| R0129 | speech: 0.719; animal: 0.656; dog: 0.647; domestic animals, pets: 0.627; bow-wow: 0.617 | birds; car; footsteps; speech; traffic | birds; car; crying; dog; footsteps; traffic |
| R0130 | speech: 0.845 | bell; car; speech; traffic | boat; car; engine; footsteps; laughter; leaves; speech; traffic; waves |
| R0131 | speech: 0.648; vehicle: 0.416 | bell; birds; car; telephone; traffic | bell; birds; car; speech; telephone; traffic |
| R0132 | speech: 0.437 | birds; traffic | birds; car; traffic |
| R0133 | speech: 0.733; vehicle: 0.518 | birds; speech; traffic | birds; footsteps; leaves; speech; traffic |

# N LLM USAGE

We used LLM to polish the text, check for typos and grammatical errors, and assist with LaTeX layout. All content was reviewed and verified by the authors, who take full responsibility for the paper's claims and conclusions.

