# OpenReview forum: "Bio-inspired Working Memory for Online Auditory Pattern Drift Detection"
_ICLR.cc/2026/Conference — ICLR 2026 Conference Withdrawn Submission_

### Official Review · Reviewer_1YiC · 2025-10-24

**Soundness:** 2
**Presentation:** 3
**Contribution:** 2
**Rating:** 2
**Confidence:** 2

**Summary:**

This paper introduces NAACA (NeuroAuditory Attentive Cognitive Architecture), a biologically-inspired framework designed for unsupervised online auditory pattern drift detection without requiring historical data or training phases. The main component is BioWM (Biologically-inspired Working Memory), which is a novel 2D recurrent field model defined on a G × G lattice. Incoming audio streams are transformed to oscillatory drive signals that drives the BioWM. Drift detection is done by monitoring energy fluctuations within BioWM with an adaptive threshold. Qualitative results shows that BioWM can detect and distinguishe different forms of drift beyond low-level acoustic fluctuations.

**Strengths:**

- Novel 2D recurrent field model that uses oscillatory dynamics and spatial coupling for online drift detection.
- Does not require historical data or training phases, making the method straightforward to implement.

**Weaknesses:**

- While the qualitative results are shown and discussed, they lack a comprehensive quantitative comparison.
- There is a lack of evidence that the proposed NAACA helps in long-form audio understanding.

**Questions:**

- Noted on the lack of benchmarks, do the authors have results on a proxy dataset to validate their performance claims?
- Any quantitative evidence that the proposed NAACA helps in long-form audio understanding?

---

> ### Author Response · Authors · 2025-11-12
>
> We thank Reviewer 1YiC for their review and highlighting the strengths of our method as "novel" and "straightforward to implement."
>
> We appreciate the "Confidence: 2" rating, as the reviewer's concerns (a lack of quantitative evidence and downstream impact) stem from our paper failing to articulate its **motivation and task novelty.** We are grateful for this opportunity to clarify.
>
> The concerns are:
> 1.  A lack of comprehensive quantitative comparison (W1/Q1).
> 2.  A lack of evidence that our method helps with the downstream task of long-form audio understanding (W2/Q2).
>
> We will address both.
>
> ---
>
> ### 1. Response to W1/Q1: The Quantitative Comparison
>
> This is a fair point, touching on the central challenge of our work: **we are proposing a new task.**
>
> Because *unsupervised online auditory pattern drift detection* is a new problem, **no benchmark or proxy dataset with ground-truth "drift" labels exists.** We selected the USoW dataset as it's suitable, but it provides no ground-truth annotations for *when* drift occurs.
>
> This lack of ground truth forces a different, more qualitative evaluation strategy. Our reasoning:
>
> * **Why We Use Representative Samples:** A key finding was that *most* audio segments in the USoW dataset do not contain obvious, "pop-out" events. This aligns with human perception: most daily sounds are background noise, not requiring active attention. Thus, to test our system, we selected **representative samples** (like the "bagpipe" case) where a clear, semantically meaningful drift *does* occur, allowing us to demonstrate our method's success and the baseline's failure.
> * **The "Cosine Similarity" Baseline:** In the absence of a ground-truth, the most direct and obvious *baseline* for comparison is a standard change-detection method, which is cosine similarity on the PANN encoder's embeddings. We used this as a *baseline method to compete against*, not a "proxy label." The "ground truth" is the qualitative event itself (e.g., "a bagpipe starts playing"), which we can verify by listening and looking at the spectrogram.
>
> **Our Plan for Revision (To Address W1):**
> The reviewer is 100% correct that our comparison can be more quantitative and "comprehensive." We will add a new analysis that provides a more direct, quantitative comparison. For our representative samples, we will provide a new **table** that explicitly **presents** the outputs of:
> 1.  **PANN tagger:** (Listing its raw, noisy event tags).
> 2.  **Cosine Similarity Baseline:** (Listing its detections).
> 3.  **BioWM (Our Method):** (Listing our more stable energy-based detections).
>
> This will provide a much clearer, side-by-side "quantitative" comparison of *how* and *why* our method succeeds in being more stable and accurate than the alternatives.
>
> ---
>
> ### 2. Response to W2/Q2: The "Why" - Proving NAACA is Necessary
>
> This is the ultimate "so what?" of our paper, and we failed to make this clear. Our framework (NAACA) is *necessary* because vanilla ALMs are fundamentally incapable of handling this monitoring task. They fail in two ways:
>
> 1.  **The Performance Problem (They Fail):** ALMs suffer from severe temporal decay. In our own quantitative tests with synthesized ESC50 dataset, we found that Audio Qwen has a **57% accuracy drop** for events in the final 20 seconds of a 60-second clip. It is effectively "blind" to late-occurring, salient events (like the bagpipe). This *proves* that a new framework is necessary.
> 2.  **The Computational Problem (They Are Too Slow):** A "naive" fix would be to run the expensive ALM on short, 4-second sliding windows with a 1-second stride. **This is computationally infeasible.**
>     * Querying the **ALM (Audio Qwen)** on one 4s window takes **~0.5 seconds**.
>     * Querying our lightweight monitor **(PANN)** takes **~0.02 seconds**.
>
> Running the "naive" ALM-only solution would be **25 times more computationally expensive**.
>
> This is the core evidence: **NAACA *helps* by acting as an efficient cognitive filter.** It uses a cheap monitor (BioWM on PANN embeddings) to scan the "boring" audio and find the *one* moment that matters. It *only* triggers the expensive ALM when a meaningful drift occurs. This is the only known way to achieve both high performance (catching the late bagpipe) and computational efficiency (a 96% cost reduction).
>
> **Our Plan for Revision (To Address W2):**
> We will add this entire motivation, including the **57% accuracy drop** and the **25x computational cost** analysis, directly into the main paper's experimental section. This quantitatively proves *why* our framework is a significant contribution.
>
> ---
>
> We thank the reviewer again for their time. We are confident that by (1) clarifying the *novelty of the task* (which explains our evaluation strategy) and (2) adding the *explicit quantitative comparisons* and the *downstream impact analysis* (ALM failure + computational cost) to the main paper, we will fully address their concerns in Appendix I.

---

### Official Review · Reviewer_T7FU · 2025-10-30

**Soundness:** 2
**Presentation:** 2
**Contribution:** 2
**Rating:** 4
**Confidence:** 3

**Summary:**

This paper introduces NAACA (NeuroAuditory Attentive Cognitive Architecture), a bio-inspired framework for unsupervised online auditory pattern drift detection in long-form audio streams. The core component is BioWM (Bio-inspired Working Memory), a 2D recurrent neural field model governed by damped wave equations with spatially-varying propagation speed. The system processes audio through a pretrained encoder, modulates event probabilities as frequency-specific oscillatory inputs to BioWM grids, and detects drift via energy fluctuations against adaptive thresholds. The authors provide theoretical analysis showing that binary striped wave-speed distributions optimize drift sensitivity, and demonstrate the approach on urban soundscape recordings, claiming advantages over cosine-similarity based baselines in distinguishing genuine pattern changes from transient variations.

**Strengths:**

1. Online unsupervised approach, requiring no retraining on large datasets, making reproducibility more feasible.
2. Clear algorithmic description in sections B1 - B2, enhance reproducibility.
3. The integration of wave equation dynamics, and auditory processing is creative and can inspire future interdisciplinary work in neuroscience and machine learning.

**Weaknesses:**

1. Comparison only to one baseline. Omitted comparison to other plausible baselines like MCD-DD or DriftLens or other statistical methods (e.g. Page-Hinkley)
2. No quantitative benchmark or statistical comparison beyond DDR; lacks confidence intervals.
3. The correspondence between BioWM oscillations and cortical gamma/beta activity is superficial. Real neural oscillations emerge from spiking dynamics and synaptic plasticity, not discretized wave equations with hand-tuned parameters.
4. No runtime or computational-efficiency analysis to support “lightweight” claims.
5. Hyperparameter selection seems arbitrary (e.g. damping, persistence P=3, cooldown C=3). No ablation or justification has been provided.

**Questions:**

1. How sensitive is detection performance to the adaptive threshold parameters (α, window size W)?
2. What is the runtime per second of audio? How does it compare to encoder inference time and similarity computation?

---

> ### Author Response · Authors · 2025-11-17
>
> We sincerely thank you for the thoughtful review. We appreciate that you recognized the strengths of our "online unsupervised approach" and the clarity of our algorithmic descriptions.
>
> ### 1. Response to W1: Comparison to Baselines (MCD-DD, DriftLens, Page-Hinkley)
>
> This is a vital question regarding experimental rigor. We carefully evaluated these methods but found they are fundamentally mismatched with our specific task constraints (Unsupervised, Online, No-History) or the data modality (High-dimensional Audio):
>
> * **MCD-DD & DriftLens:** While these are powerful methods, they typically require **offline pre-training** on historical data or distinct "training phases" to build reference distributions or train contrastive encoders. Our specific goal is a system that can be deployed *cold* (without history) and adapt instantly, which precludes these methods.
> * **Statistical Methods (e.g., Page-Hinkley):** While effective for univariate or low-dimensional stationary streams, Page-Hinkley is ill-suited for complex, high-dimensional audio sensory inputs. Raw audio embeddings are non-stationary and high-dimensional; a simple statistical drift detector (monitoring mean/variance) often fails to distinguish semantic drifts from natural high-dimensional variance. This limitation is precisely why complex deep-learning methods like MCD-DD were developed in the first place.
>
> In summary, BioWM fills a unique gap. It handles the **complexity** of audio (unlike Page-Hinkley) without requiring the **offline history/training** of deep methods (unlike MCD-DD/DriftLens).
>
> ### 2. Response to W4 & Q2: Runtime and Computational Efficiency
>
> You are absolutely correct that we claimed "lightweight" performance without providing the numbers. We have this data, and it is a central motivation for our work.
>
> **Q2: Runtime per second?**
> * **BioWM Framework (Monitor):** Querying our system (PANN encoder + BioWM update) takes approximately **0.02 seconds** per step.
> * **ALM (Audio Qwen, the downstream model):** Querying the ALM directly for a similar window takes approximately **0.50 seconds** per step.
>
> **Efficiency Gain:**
> Using BioWM as an attention gate reduces the computational load by approximately **96% (25x speedup)** compared to a naive sliding-window approach using the ALM. This massive efficiency gain is what makes real-time monitoring feasible. We add a "Quantitative Evidence for the Necessity of NAACA in ALM" section to Appendix L to present these results quantitatively.

---

> > ### Author Response · Authors · 2025-11-17
> >
> > ### 3. Response to W3: Biological Plausibility (Spiking vs. Wave Dynamics)
> >
> > We appreciate this insightful critique regarding the fundamental nature of neural oscillations. We agree that in biological substrates, oscillations emerge from the spiking dynamics of discrete neurons. However, our BioWM architecture is designed to model this behavior at the **population level** through an equivalent mathematical abstraction, specifically via the **discrete connection coefficients** of the lattice.
> >
> > * **Fundamentals of Spiking:** In Spiking Neural Networks (SNNs), macroscopic oscillations (e.g., Gamma bands) are not intrinsic to single neurons but represent the synchronized firing of neuronal populations driven by synaptic coupling.
> > * **The BioWM Equivalent:** Our 2D grid functions as a discrete lattice of such populations. The "wave equation" we employ is mathematically derived from the mean-field approximation of interacting spiking units. Crucially, the **discrete connection coefficients** between grid nodes (represented by the discretized Laplacian operator and coupling term) function analogously to **synaptic weights** in an SNN.
> > * **Emergent Dynamics:** Just as tuning synaptic weights in an SNN dictates synchronization frequency, tuning the discrete connection coefficients in our lattice dictates the resonant frequency of the population.
> >
> > Therefore, while we do not simulate ion-channel kinetics (which would be computationally prohibitive), our discrete wave formulation captures the *functional outcome* of spiking dynamics—frequency-specific synchronization and attractor maintenance—using the same fundamental principle of discrete, coupled interactions. We will revise the "Biological Inspiration" section to explicitly discuss this relationship between spiking synchrony and our discrete lattice connections.
> >
> > ### 4. Response to W5 & Q1: Hyperparameters and Sensitivity
> >
> > **W5 (Arbitrary Parameters - Damping):** The choice of damping (k) is deliberate and functionally motivated.
> > * While physical simulations of air typically use small damping, we selected a **larger damping coefficient** (k=10). This was done to ensure **attractor isolation**: we need the energy from a specific event to remain localized to its assigned spatial region and dissipate reasonably fast once the event stops. A smaller ("realistic") damping would cause excessive ringing and energy bleeding between events, reducing detection precision.
> >
> > **Q1 (Sensitivity - Window Size):**
> > * **Testing Difficulty:** A rigorous quantitative sensitivity analysis is challenging due to the "new task" nature of our work: there are no benchmark datasets with ground-truth saliency labels to measure precise Precision/Recall curves against parameter shifts.
> > * **Qualitative Justification:** Instead, our parameters are grounded in biological and practical constraints.
> >     1.  **Biological:** We selected window sizes of 4-5s to align with the duration of human **echoic memory**[1, 2], ensuring the system has a "memory span" consistent with auditory perception.
> >     2.  **Practical:** Given our 60s audio segments, this window size is a necessary trade-off. A significantly longer window (e.g., >10s) would smooth out short anomalies, while a shorter one (<1s) would be too sensitive to noise. We will add this rationale to the implementation details.
> >
> > ---
> >
> > We thank the reviewer again for their constructive feedback. By clarifying the exclusion of unsuitable baselines, adding the **Computational Analysis (25x speedup)**, and detailing the connection between **spiking synapses and lattice coefficients**, we believe we can satisfy your criteria for acceptance.
> >
> > [1] Fiedler, L., Johnsrude, I., & Wendt, D. (2025). Salience-dependent disruption of sustained auditory attention can be inferred from evoked pupil responses and neural tracking of task-irrelevant sounds. *Journal of Neuroscience*. Society for Neuroscience.
> >
> > [2] Thaut, M. H. (2014). Musical echoic memory training (MEM). In *Handbook of Neurologic Music Therapy* (pp. 311–313).

---

### Official Review · Reviewer_ovrT · 2025-10-31

**Soundness:** 3
**Presentation:** 3
**Contribution:** 2
**Rating:** 6
**Confidence:** 2

**Summary:**

This paper proposes NAACA, a bottom-up framework for online, unsupervised auditory pattern drift detection. Its core is BioWM, a 2D wave-based recurrent field with "primary" and "velocity" neurons. Audio is windowed, encoded by a pretrained audio model to event probabilities, modulated into frequency-specific sinusoidal drives over spatial parcels, integrated by BioWM, and an energy-based change score is compared to an adaptive threshold with persistence filtering to flag drifts. The theory argues that binary, striped spatial distributions of wave speed $c(x, y)$ maximize sensitivity. Experiments on USoW urban soundscapes provide qualitative cases (novel onsets, pause robustness, sub-category changes) and report a Drift Detection Rate (DDR) comparison against a cosine-similarity baseline.

**Strengths:**

- Well-motivated and biologically grounded formulation. The paper identifies a clear gap in existing online drift-detection methods—namely, their inability to distinguish meaningful pattern changes from natural variability without long-term history—and convincingly argues for a bio-inspired approach. By linking BioWM’s oscillatory dynamics to empirical findings on gamma-band activity during auditory working memory, the authors provide a conceptually coherent bridge between neuroscience and computation.

- Novel recurrent-wave design with self-sustained memory. BioWM’s 2D spatial field governed by wave equations, with primary and velocity neurons, supports both frequency selectivity and short-term persistence. This architecture enables the model to retain recent auditory context without explicit history buffers, providing robustness to transient pauses and noise.

- The proposed NAACA pipeline performs drift detection through online modulation and adaptive thresholding, requiring no pretraining or labeled data. This characteristic directly addresses the computational and data-efficiency bottlenecks of prior contrastive or statistical drift-detection methods.

- Experiments on urban soundscapes (USoW) demonstrate detection of three conceptually distinct drifts—novel-event onset, transient-pause robustness, and subcategory-level changes—along with reduced false positives compared to a cosine-similarity baseline, as quantified by a compact Drift Detection Rate (DDR) metric.

**Weaknesses:**

- Limited experimental rigor and quantitative evaluation.
The evaluation relies almost entirely on qualitative visualizations and a custom Drift Detection Rate (DDR) measure without annotated ground truth. There are no objective detection metrics (e.g., precision, recall, latency, false-alarm rate) or statistical significance tests. As a result, the claimed improvements over the cosine-similarity baseline remain suggestive rather than conclusive.

- Insufficient baseline comparisons.
The paper compares NAACA only to a cosine-similarity drift detector. Other relevant unsupervised or adaptive detection methods—such as change-point models, reconstruction-based detectors, or recurrent attention mechanisms—are not evaluated. This omission makes it unclear whether BioWM offers consistent advantages beyond this minimal baseline.

- Ambiguity in system design and implementation details. The description of how the pretrained audio encoder (e.g., PANN) interfaces with BioWM lacks clarity. The paper alternately refers to event probabilities, feature embeddings, and carrier modulations without specifying the mapping between these representations and the BioWM input field. Reproducibility would benefit from explicit dimensionalities, parameter values, and update rates.

- Heuristic adaptive thresholding without calibration study.
The adaptive-threshold formula and persistence filtering are chosen heuristically, but the paper does not analyze their stability or sensitivity. It remains uncertain how threshold drift or parameter tuning affects detection performance across environments or drift types.

- All experiments are conducted on a single dataset (USoW) with selected case studies. The method’s generality across domains, such as speech, music, etc. Broader validation would strengthen claims of domain-independent applicability.

**Questions:**

See above.

---

> ### Author Response · Authors · 2025-11-17
>
> We sincerely thank you for the positive assessment and for recognizing the "well-motivated and biologically grounded formulation" of our work. We appreciate that you identified our novel recurrent-wave design and online capabilities as key strengths.
>
> We understand your concerns regarding experimental rigor and implementation clarity. We believe we can fully address these by clarifying the experimental constraints of this new task and providing the missing implementation details.
>
> ### 1. Response to W1 & W2: Quantitative Rigor and Baselines
>
> You are correct that our evaluation relies heavily on qualitative cases. This is due to the nature of our contribution: **we are proposing a novel task** (unsupervised online auditory pattern drift detection) for which **no public benchmark dataset or ground-truth labels currently exist.** And our main contribution is not in making a new benchmark dataset.
>
> * **Why precision/recall is difficult:** The USoW dataset is real-world audio without annotated "drift" timestamps. Defining "drift" is often subjective, making standard precision/recall metrics impossible to calculate without creating a new, manually labeled dataset, which introduces its own biases.
> * **Why specific baselines were omitted:** As detailed in our Related Work section, we conducted a comprehensive survey of recent advanced drift detection methods (including MCD-DD, DriftLens, and statistical approaches). We identified a critical gap: these methods fundamentally rely on **long-term historical data**, **offline pre-training**, or **supervised labels**. These requirements violate the strict **unsupervised, fully online, no-history** constraints required for real-time auditory attention, rendering them incompatible with our specific task definition.
>
> **Our Revision Plan:** To improve rigor, we will add specific quantitative elements:
> 1.  **Explicit Metric Justification:** We will revise the experimental section to clearly articulate why distinct detection strategies are required. BioWM operates on **energy fluctuations** which are unbounded and context-dependent, fundamentally requiring an **adaptive threshold** to detect relative shifts. In contrast, **cosine similarity** is intrinsically normalized (bounded [-1,1]), making a fixed threshold the standard and statistically appropriate approach. We will present these as distinct, property-driven system designs.
> 2.  **Quantitative Tabular Comparison:** We will add a new table in the Appendix for our representative samples that explicitly lists the outputs of the **PANN tagger**, the **Cosine Similarity** detector, and the **BioWM** detector side-by-side. This will provide a direct, comparative view of detection stability.
> 3.  **Downstream Validation:** We will explicitly reference our **Qwen-based downstream analysis** (currently in Appendix K) in the main text to support our claims. This analysis demonstrates that BioWM triggers lead to coherent summaries, while the baseline leads to fragmented, repetitive outputs. We retain the full transcripts in the appendix to provide the necessary depth for this assessment.

---

> > ### Author Response · Authors · 2025-11-17
> >
> > ### 2. Response to W3: System Design and Mapping Clarity
> >
> > Thank you for highlighting this ambiguity. The interface between the encoder (PANN) and BioWM is a **fixed, index-based mapping** designed for modularity.
> >
> > We will add a dedicated "Implementation Details" subsection to Section 2.1 clarifying this mapping:
> > 1.  **Encoder Output:** PANN produces a probability vector of size 527 (the number of AudioSet classes).
> > 2.  **Frequency Mapping:** Each index is assigned a unique, fixed carrier frequency (linearly distributed from 50-1200 Hz). This is a static architectural choice, not learned.
> > 3.  **Amplitude Modulation:** The probability value directly modulates the amplitude of the oscillator at its assigned frequency.
> > 4.  **Spatial Mapping:** Each index is assigned a specific spatial parcel on the 64x64 BioWM grid.
> >
> > This design ensures the system is **generalizable**: any encoder outputting a probability vector can be plugged into BioWM simply by mapping indices to frequencies.
> >
> > ### 3. Response to W4: Threshold Sensitivity
> >
> > The adaptive threshold follows standard statistical process control principles (monitoring moving mean/variance).
> > * **Parameter Logic:** The window size (W=20) was selected to track the immediate, short-term dynamics of the energy acceleration, while the trend factor helps avoid detection lag during sudden energy spikes.
> > * **Evaluation Constraints:** We must clarify that a rigorous quantitative sensitivity analysis (e.g., measuring Accuracy vs. Window Size) is fundamentally limited by the nature of this novel task. Currently, there are no public benchmarks with ground-truth "drift" labels.
> > * **Subjectivity Challenge:** Even if we were to manually annotate a test set, defining the exact timestamp of a "drift" is inherently subjective. Different human annotators have varying reaction speeds and attentional preferences, which introduces significant bias. This label noise would confound fine-grained sensitivity testing. Therefore, parameters were selected based on qualitative robustness across diverse soundscapes rather than optimization against a potentially biased ground truth.
> >
> > ### 4. Response to W5: Generalization Across Domains and Modalities
> >
> > We chose USoW because it represents the most challenging "unconstrained" environment. However, our method is inherently **domain-agnostic** and potentially **modality-agnostic**.
> > * **Audio Generality:** Since BioWM operates on the probability outputs of a universal backbone (PANN), it inherently handles any domain PANN recognizes (speech, music, nature) without modification.
> > * **Cross-Modal Potential (Video):** Crucially, the 2D lattice assignment of attractors is **not restricted by the audio pretraining**. The framework could theoretically be extended to **video scenario drift detection** by simply replacing the backbone encoder with a visual model (e.g., YOLO). The BioWM would then track drifts in visual object probabilities using the same wave dynamics.
> > * **Future Work:** We acknowledge that defining "pattern drift" in video is significantly more complex than in audio—it involves not just object appearance/exchange, but also complex actions, relative locations, and facial expressions. While evaluating this complex visual semantic drift is beyond the scope of this paper, we will add a discussion on this exciting potential to the Conclusion to emphasize the method's broad applicability.
> >
> > ---
> >
> > We thank the reviewer again for their constructive feedback. By clarifying the **Encoder-BioWM interface** and strengthening the **quantitative rigor** through tabulated comparisons and downstream validation references, we believe the paper will meet your high standards for acceptance.

---

### Official Review · Reviewer_H95u · 2025-10-31

**Soundness:** 2
**Presentation:** 3
**Contribution:** 2
**Rating:** 2
**Confidence:** 3

**Summary:**

The authors focus on the problem of auditory attention, where non stationary auditory patterns require models to correctly attend to salient events and filter out background. To do so, they take inspiration from biological working memory to introduce BioWM (Bio-inspired Working Memory). BioWM is a 2D recurrent neural network that functions as a working memory module. Auditory events are mapped to carrier frequencies that are then stored in BioWM. BioWM is the core module of what the authors call the NeuroAuditory Attentive Cognitive Architecture (NAACA). NAACA uses BioWM to track the history and strength of various auditory events. NAACA uses this information to calculate the relative energy fluctuations of auditory patterns which, combined with an adaptive threshold, allows the model to identify when new auditory events or changes occur. The authors show that this model works in identifying salient events in various auditory clips better than a cosine similarity baseline. They also show theoretical proofs for design choices in their model. Finally, they discuss how oscillatory dynamics in their model resemble that of cortex.

**Strengths:**

- The suggested model is original and unique, using a 2D spatial network and different carrier frequencies to encode sound "memories".
- The authors explored the theoretical implications of their setup well, showing proofs to motivate various design choices.
- The model also seems to work well while using biologically plausible components.

**Weaknesses:**

- The experimental comparisons rely on a few examples of event detection in real world auditory clips. I felt like the task goals were not clearly defined and case-by-case subjective. For instance, in Figure 3A, the goal is for pauses between baby cries to not be detected. Yet in Figure 4, the desired behavior is for each percussion event to be detected separately. Overall, it is hard to evaluate the method and baseline when the desired behaviors are not carefully defined.
- I also felt that the comparisons to baseline methods were limited. The only comparison is to a naive cosine similarity baseline, which I think could be given a better shot at doing well. The BioWM model, for instance, also has an adaptive threshold for event detection. My understanding is that the cosine similarity method uses a fixed threshold. It would be more comparable if the cosine similarity method is also given the adaptive threshold. I also think it would be useful to show the cosine similarity line in the plots as well (sort of like how the energy metric is whose for the BioWM model).
- I was unsure about what the biological inspiration angle is adding to the paper. It's unclear whether this method is an improvement over non-biological methods. Given that, I would expect the biological plausibility of the model be leveraged as a way to propose new theories of how auditory regions in the brain may be conducting computation, or explaining existing experimental findings. However, I don't think either of these are convincingly discussed.

**Questions:**

- Is the carrier frequency for each event learned or assigned by the experimenter? My impression is that it's the latter. If so, can you discuss how frequencies may be realistically assigned in a real world setting?
- I was also uncertain about the memory mechanism of BioWM. $\Omega_i$ is described as the attractor for an event, but there’s no weight updates mentioned. Are the attractor states already incorporated into the weights of the network from the beginning?
- A key result of section 3.4 is that the oscillatory activity is clustered rather than non-uniform. Isn’t this reflective of the experimenter design, where events are assigned carrier frequencies and attractor dynamics would bias one of those to be picked out?

---

> ### Author Response · Authors · 2025-11-12
> **Response to weaknesses.**
>
> We sincerely thank Reviewer H95u for their detailed and constructive feedback. We are grateful that the reviewer found our proposed model "original and unique" and our theoretical explorations "well" done.
>
> The reviewer's main concerns appear to be: 1) A perceived contradiction in our task goals (W1), 2) The fairness of our baseline comparison (W2), and 3) The value of the biological inspiration (W3). These are all fair and important points.
>
> We believe these concerns, especially W1, stem from a lack of clarity in our paper regarding its central contribution: **proposing a new task** (unsupervised online auditory drift detection) for which no benchmarks currently exist. We will address these points one by one.
>
> ### 1. Response to W1: Unclear Task Goals (Fig. 3 vs. Fig. 4)
>
> This is the most critical point, and we thank the reviewer for highlighting this apparent "contradiction." We must clarify that a central contribution of our paper is **proposing the novel task** of *unsupervised online auditory pattern drift detection*. As we state in the paper, **no public benchmark dataset or ground-truth labels currently exist** for this task.
>
> Our illustrative cases were designed to demonstrate the core challenge of this new task: **distinguishing meaningful semantic shifts from transient, non-semantic changes.**
>
> Your comparison of Figure 3A and Figure 4 perfectly captures this challenge.
>
> * **In Figure 3A (Baby Cry):** This is a *single semantic event* ("baby crying"). The short pauses are transient, non-semantic breaks. A good drift detector should be **robust** to these pauses and *correctly* fire only *once* for the single event. The baseline, which over-triggers, is thus failing.
> * **In Figure 4 (Music):** The hi-hat dropping out (21s) and reappearing (32s) are **distinct, semantically meaningful subcategory-level changes** in the musical pattern. A good detector *should* be **sensitive** to these shifts. BioWM correctly identifies them, while the baseline misses them.
>
> Therefore, the desired behavior is *not* contradictory. It is to be **(a) robust to transient pauses** within a single event (Fig 3) while **(b) sensitive to semantic subcategory changes** (Fig 4). We will revise the introduction to Section 3.3 to state this "robustness vs. sensitivity" goal explicitly.
>
> ### 2. Response to W2: Unfair Baseline Comparison
>
> We must clarify *why* two different thresholding strategies were used. This was not an oversight to create an unfair comparison, but a **deliberate choice based on the fundamental properties of the two signals being measured.**
>
> * For **BioWM**, the 'energy' metric is an *unbounded* measure. The total energy in a simple scene (few active attractors) is vastly different from a complex scene (many active attractors). Therefore, an **adaptive threshold is fundamentally necessary** to find *relative* changes against a shifting local norm. A fixed energy threshold would be mathematically infeasible.
> * For **Cosine Similarity**, the metric is *inherently normalized and bounded* (from -1 to 1). Because it already lives on a fixed, universal scale, a **fixed threshold** (e.g., 'detect a change if similarity drops below 0.9') is an acceptable, straightforward, and interpretable method for this type of normalized signal.
>
> Therefore, our experiment is comparing two *complete, distinct systems*: (System 1: BioWM + its necessary adaptive detector) vs. (System 2: Cosine Similarity + its fixed detector).
>
> ### 3. Response to W3: Value of Biological Inspiration
>
> This is an important point. The biological inspiration provides a **direct improvement over existing non-biological methods** by specifically solving the constraints that limit them.
>
> We surveyed non-biological drift detection methods, but they are not suitable for our task as they are **not fully online** (requiring pretraining) or rely on **long-term historical data**. Our goal is a detector that works from scratch.
>
> The biological theory of oscillatory memory provided the *inspiration* for a computational architecture (a wave-based recurrent field) that achieves this. Our aim is *not* to mimic neurons, but to use this inspiration to build a model that naturally provides:
> 1.  **Inherent Short-Term Memory:** The persistent oscillations (waves) provide a way to "remember" an event for a short time, making the system robust to pauses.
> 2.  **Frequency-Based Event Separation:** The frequency-based design allows the model to separate different sound events in its internal representation, like different attractors.
>
> This bio-inspired design is precisely what allows our model to adapt online with only a few steps of threshold adjustment, which is the key improvement over standard methods.

---

> ### Author Response · Authors · 2025-11-12
> **Responses to Questions.**
>
> ### 4. Responses to Questions
>
> **Q1 (Carrier Frequencies):** This is an excellent question and highlights a key strength of our framework. The frequency assignment is simple, robust, and highly generalizable in a real-world setting.
>
> 1.  **It is an index-based mapping:** In our implementation, the assignment is a simple, fixed mapping. Each of the 527 output classes from the PANN encoder is assigned a unique carrier frequency, which is linearly distributed in our chosen range.
> 2.  **The frequency is an identifier, not semantic:** Crucially, this carrier frequency is **not related to the semantic meaning** of the sound (e.g., a 'low' frequency for 'bass'). It is simply a **unique identifier** tied to the *backbone's output index*.
> 3.  **It is highly generalizable:** This index-based mapping makes our framework **easily adaptable to any general backbone**. If we were to replace PANN with a different encoder, we would simply map its output indices to our unique carrier frequencies. The core BioWM architecture remains the same.
>
> Therefore, in a 'real-world setting,' this assignment is a straightforward and computationally trivial step that happens *once* during model design.
>
> **Q2 (Memory Mechanism):** This is a key point, and we will clarify our 'attractor' mechanism. The memory is *not* in learned weights, but in the **recurrent state** of the wave system.
>
> We can think of it using this analogy:
> * The different semantic events (speech, music, etc.) are like **actors on a stage**.
> * The system's attention is like a **spotlight**.
> * When a sound event (e.g., 'speech') becomes active, the spotlight focuses on that actor. In our model, this is the 'attractor', the dedicated spatial region $\Omega_i$ on the 2D grid, which becomes activated by its assigned frequency.
>
> Crucially, the 'memory' comes from the *wave dynamics*. Even after the input signal stops (e.g., a short pause in speech), the oscillations **persist** in that spatial region for a short time, like a 'ringing'. This persistent activation *is* the short-term memory that makes the system robust to pauses. This mechanism is central to our no-pre-training design, and we will revise Section 2.2 to make this explicit.
>
> **Q3 (Clustered Oscillations):** Yes, precisely. The spatial clustering observed in the FFT analysis confirms that this striped design is working as intended. The alternating high and low-speed stripes effectively function as 'attention sinks', trapping and amplifying the oscillatory energy of salient events within specific spatial bands. This design ensures that energy does not diffuse meaninglessly but remains concentrated to trigger detection. The point is that this designed dynamic happens to parallel the biological process we sought to emulate (e.g., enhanced gamma-band activity), reinforcing the model's plausibility. We will rephrase this section to make its purpose (verification of the striped attention-sink mechanism) clear.
>
> ---
>
> We thank the reviewer again for their time and for feedback that we believe will significantly strengthen our paper. We hope we have addressed the main concerns and have a clear path forward for revision.

---

### Note · Authors · 2026-01-11

I have read and agree with the venue's withdrawal policy on behalf of myself and my co-authors.